# Cross-platform clinical proteomics using the Charité open standard for plasma proteomics (OSPP)

Ziyue Wang [1], Vadim Farztdinov[2], Ludwig Roman Sinn [1], Pinkus Tober-Lau [3], Daniela Ludwig[1,2], Anja Freiwald[1,2], Fatma Amari[1,2], Kathrin Textoris-Taube[1,2], Agathe Niewienda[1,2], Anna Sophie Welter [4,5], Alan An Jung Wei[4], Luise Luckau[6], Florian Kurth [3], Matthias Selbach [4], Johannes Hartl [7], Michael Mülleder [2,10] ✉ & Markus Ralser [1,7,8,9,10] ✉

The role of plasma and serum proteomics in characterizing human disease, identifying biomarkers, and advancing diagnostic technologies is rapidly increasing. However, there is an ongoing need to improve proteomic workflows in terms of accuracy, reproducibility, and cost-effectiveness, and to achieve cross-platform transferability. Based on large serum and plasma proteome studies, we generate the Charité *Open Peptide Standard for Plasma Proteomics* (OSPP), an open, versatile peptide internal standard for targeted and untargeted mass spectrometry-based proteomic studies. The OSPP includes 211 concentration-matched stable-isotope-labeled peptides selected for consistent quantification across a large number of plasma and serum proteome studies, and synthetic accessibility. We show they are consistently quantified across serum and EDTA, citrate, and heparin plasma using multiple LC-MS platforms. Despite being selected for technical parameters, the OSPP peptides represent proteins that function in a wide range of biological processes, are used in routine clinical tests, or are targets of FDA-approved drugs, making OSPP able to serve as an expandable clinical marker panel. We demonstrate the utility of OSPP in a COVID-19 inpatient cohort study for improving analytical performances, for cross-platform alignment of proteomic data, disease stratification, and biomarker discovery.

The analysis of human plasma and serum proteomes is of increasing importance for biomedical research due to the minimally invasive sample collection of plasma and serum combined with the properties of the plasma proteome to directly reflect human physiological and/or pathophysiological states[1–4]. Plasma proteomes reflect various physiological processes, and plasma proteins can function as biomarkers for different conditions. This includes well-established biomarkers such as cardiac troponin T, C-reactive protein, procalcitonin, and cystatin C, supporting decision-making related to diagnosis, prognosis, and intervention monitoring[5–9].

Liquid chromatography-mass spectrometry (LC-MS) allows the simultaneous quantification of large numbers of proteins, elucidating the functional implications of specific proteins and unveiling changes within pertinent pathways while maintaining relatively low operational costs[3,10–12]. Nevertheless, the routine implementation of LC-MS, especially discovery proteomics, still faces challenges arising from technical variances in acquisition batches and methods, analytical platforms, and processing pipelines. The presence of these variations results in incomparable relative quantification of proteins across platforms, adding complications to the data

analysis and the routine implementation of the proteomic workflows.

A strategy for addressing these challenges is to include different types of analytical standards and to aim for absolute rather than relative quantification methods. In mass spectrometry, stable isotope-labeled internal standards (SIL-IS) have become increasingly popular since the 1990s and allow the normalization of endogenous signals with absolute reference values[13,14]. In bottom-up proteomics, the spike in SIL-IS with $^{13}$C, and $^{15}$N labels is commonly used[15–19], and serves for quality control procedures, batch correction, retention time normalization, and for estimating absolute quantities at the peptide level. Further, these standards can help to achieve cross-platform and cross-laboratory reproducibility[16,20–22]. Commercially available peptide standards, such as the PQ500 standard (Biognosys, Switzerland)[23–25] or the PeptiQuant 270-protein human plasma MRM panel (MRM Proteomics Inc., Canada)[26,27], are designed for targeted analysis of broad sets of proteins. However, there is a continuing need to optimize peptide selection for constant peptide detection across platforms, studies, and matrices; for costs per sample; and to improve data analysis strategies to efficiently make use of internal standards not only in targeted but also in discovery proteomics.

In this work, to respond the above mentioned challenges, we extensively mined a large number of plasma and serum proteomic studies to select peptides with ideal properties for a broadly applicable and expandable SIL-IS and generated the Charité Open Peptide Standard for Plasma Proteomics (OSPP). Consisting of 211 isotope-labelled peptides, the OSPP peptides exhibit high analytical performance across various analytical platforms and blood matrices. Additionally, they are selected to be easily and economically synthesized and be broadly applicable across various clinical contexts. Moreover, we report their constant detection at low technical variance across platforms. As the OSPP peptides are derived from proteins involved in a wide range of biological processes, many of which are targets of FDA-approved drugs or routinely used biomarkers, quantifying the OSPP peptides directly captures biological responses and can thus be used as a basis and expanded into biomarker panels.

## Results

### Selection of analytically robust peptide standards from large-scale proteomic data sets

To identify peptides with ideal properties to be combined into an SIL-IS standard panel addressing neat serum and plasma proteomics, we exploited proteomic data produced in 1505 control injections, which were measured alongside the analysis of 15,617 plasma and serum proteomes at the Charité High-throughput Mass Spectrometry Core Facility from 2020 to 2022. Data was analyzed using DIA-NN[28] using a publicly avalaible spectral library (DiOGenes study)[29].

To prioritize the most reliably quantified precursors from tryptic peptides that are proteotypic to the respective protein, we introduced a relative rank metric. This metric was calculated by first defining the precursor weight as the ratio of the precursor's detection frequency/ percentage of presence ($PPres(p,n)$) to its coefficient of variation ($CV(p,n)$) evaluated on an intensity scale. The weight of the precursor thus corresponds to its signal-to-noise ratio ($S/N = 1/CV$), then multiplied by its presence.

$$Weight(p,n) = PPres(p,n)/CV(p,n). \qquad (1)$$

Here, $p$ stands for precursor and $n$ for a study pool series. Further, the rank of a precursor is assigned by its weight. Here and below, the index of a function indicates the variable to which the function is applied.

$$Rank(p,n) = rank_p\{Weight(p,n)\}. \qquad (2)$$

The number of precursors was different between studies, ranging from 4653 to 6361, with an average of 5390 precursors in all studies (Supplementary Data 1−selection cohort). To account for differences in the total number of precursors across studies, we calculated a relative rank ($RelRank(p,n)$) for each precursor in each study, defined as the rank of a precursor ($Rank(p,n)$) divided by the maximum rank observed in that study ($\max_p\{Rank(p,n)\}$).

$$RelRank(p,n) = Rank(p,n)/\max_p\{Rank(p,n)\} \qquad (3)$$

To select the best-performing, non-project-specific precursors for every protein, we calculated the average (over considered studies) relative rank $RelRank(p)$ and set the lower cutoff as 0.5.

$$RelRank(p) = mean_n\{RelRank(p,n)\} \qquad (4)$$

$$(RelRank(p) \geq 0.5) \qquad (5)$$

The proteotypic peptides from the best-performing, non-project-specific precursors were further selected to be quantified in at least half of the examined studies and covered at least two blood matrices.

$$Proteotypic(p) = 1, PresInNProjects(p) \geq 4, PresInNMatrices(p) \geq 2 \qquad (6)$$

To allow coverage of a larger protein concentration range, not more than three top-ranked peptides were selected for each protein (Fig. 1). Eventually, this selection process identified 382 consistently quantified precursors. We further selected these according to their suitability as internal standards, such as a peptide length between 6-25 amino acids, a minimal likelihood of missed cleavages, and low susceptibility to chemical modifications (Fig. 1). Among these, we prioritized peptides with high synthesis efficiency according to the Peptide Analyzing Tool (Thermo Scientific)[30]. We also incorporated 24 out of 50 peptides from a previous peptide panel, which showed excellent cross-platform analytical performances[31,32].

Eventually, this selection procedure converged on 187 new peptides, expanded by 24 peptides that were already part of a previous SIL-IS panel designed to assess disease severity in COVID-19 and MPox patients[31,32]. These 211 peptides are derived from 131 proteins, of which 71 proteins are represented by one internal standard peptide, 44 proteins are represented by 2 internal standard peptides, 13 proteins by 3 internal standard peptides, 2 proteins, APOA1 and APOB, represented by 4 internal standard peptides, and SERPINA3 is represented by 5 internal standard peptides (Supplementary Data 1−peptide properties, Supplementary Fig. 1). These proteins span over four orders of magnitude in the concentration range of human plasma (Fig. 2a). Moreover, they have been found to be associated with several disease types (Fig. 2b, upper panel), encompassing enzymes, transporters, and cytokines (Fig. 2b, middle panel), and cover a broad range of 35 FDA-approved drug targets[33]. Notably, a fraction of these proteins, selected for purely technical reasons, already serve as routine clinical chemistry biomarkers used in different matrices (serum, citrate-, heparin-, or EDTA-plasma) (Fig. 2b, lower panel).

### Assessment of selected peptide standards

The selected peptides were synthesized in both unlabeled (native) and isotopically labeled (SIL) forms, with heavy labeling of the C-terminal arginine or lysine using $^{13}$C$^{15}$N. Validation of the synthesized peptides involved analysis by LC-UV/VIS by the peptide manufacturer (Pepmic, Inc.) and LC-MS analysis. Subsequently, unlabeled and SIL peptides were prepared in groups of 11 based on their abundance in EDTA plasma and analyzed with a 5 μl/min microflow-rate reversed-phase chromatographic gradient by Zeno SWATH DIA on a ZenoTOF 7600

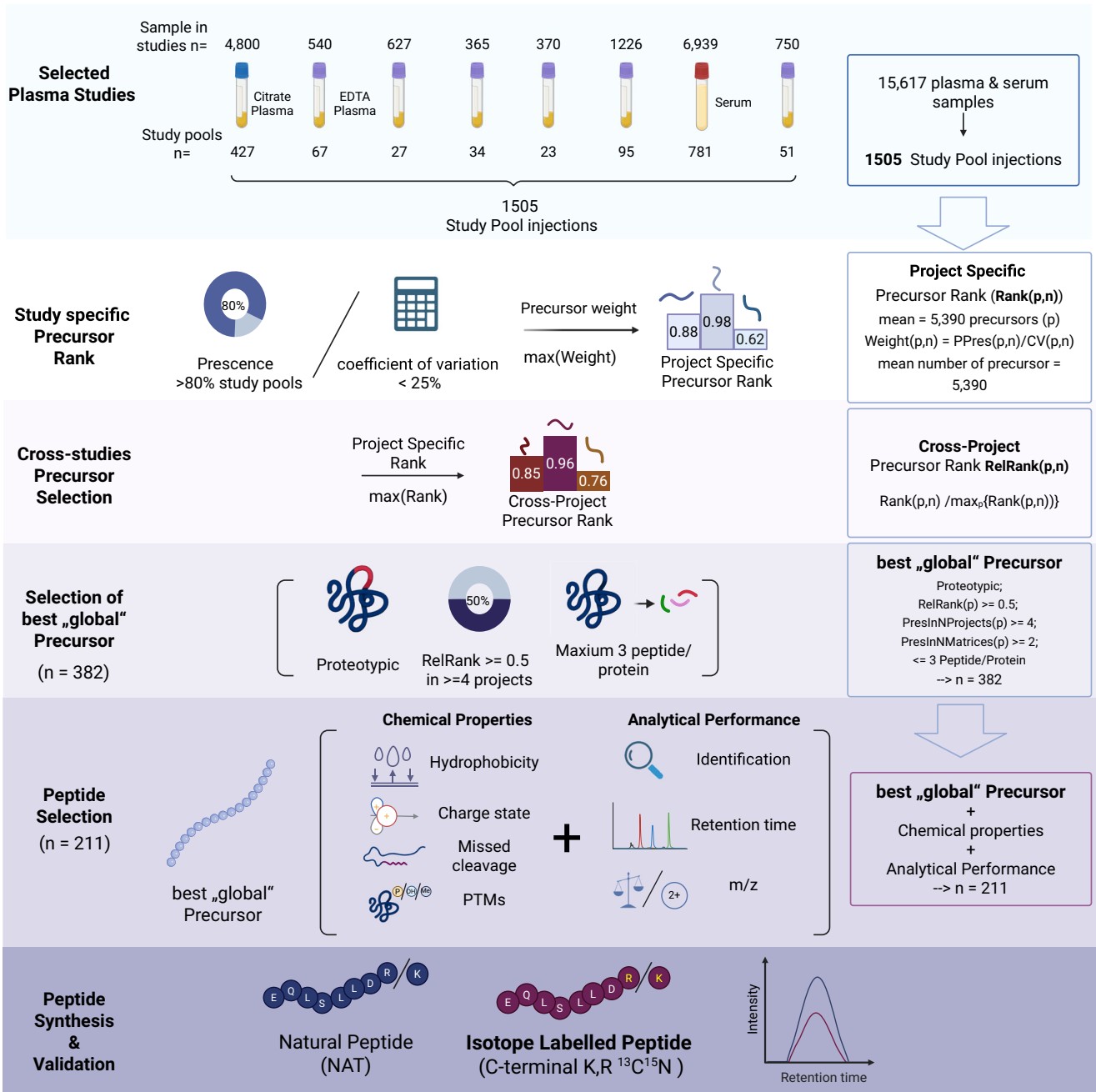

**Fig. 1 | Overview of the peptide selection process to generate the Charite open standard for plasma proteomics (OSPP).** Plasma and serum proteomic data acquired from 1505 measurements of study control samples were used for selecting 187 peptides with ideal properties of an internal standard (Scheme, Methods). Peptide selection for the OSPP was based on consistent detection across studies, their signal stabilities, chemical and biophysical properties, proteotypicity, and suitability for chemical synthesis. In addition, we added 24 peptides out of a previously generated targeted peptide panel assay[31] The total of 211 selected peptides (Supplementary Data 1−peptide properties) were synthesized in both native and isotopically labeled form. Created in BioRender. WANG, Z. (2025) https://BioRender.com/8az8cdz.

instrument (SCIEX)[34]. Quality criteria entailed the co-elution of SIS with their respective unlabeled forms and the absence of signals from the unlabeled standard peptides. Reassuringly, the selected peptides all have well-distributed hydrophobicity scales, achieving a balanced elution distribution across the entire retention time range in a 20 min active gradient microflow chromatography (Fig. 2c, Supplementary Data 1−peptide properties).

Next, we tested the analytical performance of the SIL-IS peptides in Heparin-, Citrate-, and EDTA-plasma, as well as in serum samples obtained from healthy volunteers (Supplementary Data 4−metadata for 4 matrix). As an initial analysis, we added an equal concentration of

each SIL-IS peptide ("Single-conc. Standard", 100 pg per peptide, 21.1 ng total peptide amount) into 1.5 µg of total protein digest from each matrix, analyzed using analytical flow rate chromatography (500 µl/min, 5 min gradient with a timsTOF HT mass spectrometer (Bruker) and analyzed using diaPASEF[35]. Despite inherent abundance changes e.g., due to age and sex differences (Supplementary Fig. 2), 93% (197/211) of peptides (both endogenous and SIL-IS peptides) were consistently detected in over 66% of samples. Moreover, 147 endogenous peptides were consistently detected in all four blood matrices, 174 in at least three matrices, and 9 endogenous peptides can be identified in a single matrix only (Supplementary Data 1−peptide

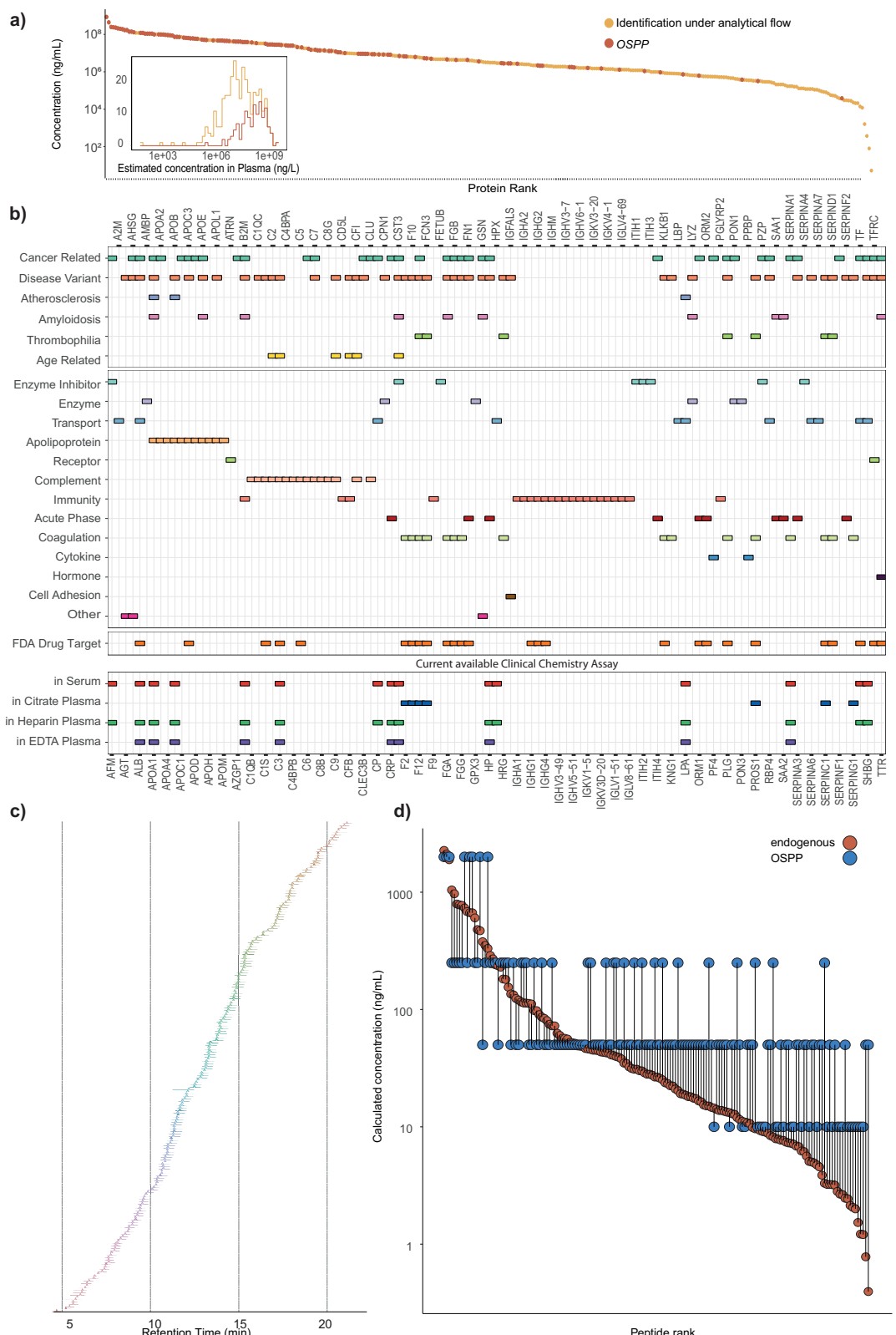

properties, Peptide quantities in different matrices; Supplementary Fig. 2b).

To further assess peptide-specific abundance differences across matrices, we performed pairwise Wilcoxon signed-rank tests to evaluate rank shifts of the respective endogenous peptides between sample matrices (Supplementary Data 1−pairwise Wilcoxon signed-rank tests). A subset of peptides maintained consistent relative abundance, with

serum and heparin plasma showing the highest similarity−148 out of 188 peptides did not change rank significantly (*p.adjust* > 0.05). In contrast, rank shifts were observed across other matrix comparisons. The largest difference was seen between serum and citrate plasma, where 143 out of 194 peptides changed rank (*p.adjust* < 0.05).

To account for the large concentration differences of proteins in plasma, we adjusted each selected peptide to align within one order of

**Fig. 2 | Characteristics of the selected peptide standards (Source data are provided as a Source Data file). a** Detection trends for proteins of the plasma proteome in analytical flow-rate chromatography, ranked by their estimated concentration on an absolute scale. Plasma proteins are represented by the selected peptides[33,78]. The inset shows the log-scale plasma protein concentration distribution of OSPP peptides compared to those identified in the analytical flow DIA dataset. The peptides are derived from proteins whose concentration spans over four orders of magnitude. Data for protein concentration was downloaded from the Human Protein Atlas (filtered to proteins detected in human plasma by MS), and albumin concentration (40 g/l) was added manually. Only the reported protein concentration was used for plotting[79]. **b** The selected proteins and their associated biological functions (upper panel), whether they are FDA-approved drug targets (middle panel) (top and middle panel categories curated with The Human Protein Atlas[33,78] in November 2023), as well as their use in routinely used clinical tests (provided by the Institute of Laboratory Medicine, Munich, Germany) based on serum, citrate, heparin, or EDTA plasma, respectively (lower panel). **c** Retention time distribution of the selected peptides in a 20 min μflow reversed-phase liquid chromatography, as analyzed by Zeno SWATH DIA on a ZenoTOF 7600 instrument (SCIEX). MS2 intensities from extracted ion chromatograms were normalized to the maximum intensity of the respective peptide peak. The OSPP peptides distribute evenly across the chromatographic gradient. **d** Dynamic range and concentration matching of OSPP peptides to endogenous plasma levels. The dynamic concentration range of OSPP peptides is based on calibration curve-derived concentrations measured in pooled EDTA plasma using microliter flow-rate chromatography and Zeno SWATH DIA on a ZenoTOF 7600 instrument (SCIEX). OSPP peptide concentrations were adjusted to match endogenous concentration levels within one order of magnitude.

magnitude of its endogenous counterpart in the pooled EDTA plasma and created a concentration-matched internal standard (Fig. 2d). Peptides were grouped into four concentration bins ranging from 10 pg/μl to 2 ng/μl (total of 40.4 ng across 211 peptides in 10% acetonitrile; Supplementary Data 2). This optimized, concentration-adjusted internal standard is referred to as the Open Standard for Plasma Proteomics (OSPP). Of note, such concentration adjustment does not only improve quantitative accuracy, as the standard and analyte are in a similar concentration range, and as the spiked peptide amount does not exceed endogenous levels substantially; it also saves synthesis costs, as low quantities are required for the low-abundance peptides.

To assess the suitability of OSPP across acquisition platforms, we generated triplicate eight-point external calibration curves for all 211 peptides, using a bovine serum albumin (BSA) tryptic digest as a surrogate matrix. The native peptide standard in all calibration curves spanned four orders of magnitude of concentration, from 0.5 ng to 0.032 pg. All detected OSPP peptides showed a good linearity with more than four calibration points, and endogenous peptide concentrations in pooled EDTA plasma all fell within this range. We also reported limits of detection and limits of quantification (lower and upper: LLOQ and ULOQ) of each peptide from each acquisition method in Supplementary Data 3.

We further compared the analytical performance of the OSPP peptides against peptides present in a commercial internal standard panel, the PQ500 (Biognosys), which contains 804 peptides derived from 572 proteins[23]. 81 peptides overlap between the standards. In pooled EDTA plasma, analyzed by micro-flow LC and Zeno-SWATH-MS in triplicate. All endogenous peptides matching the 81 overlapping peptides are detected and quantified with a median coefficient of variation (CV) of 5.45%. For proteins included in both panels but represented with different peptide selections, OSPP-specific peptides ($n = 108$) showed a median CV of 6.12%, while the PQ500-specific peptides, which quantified the same proteins ($n = 133$), had a higher median CV of 7.31%. Overall, 98% of OSPP peptides (207/211) were consistently quantified with a median CV of 6.11%, while the PQ500 panel contains more peptides, but only 352 (43%) were consistently detected in our setup, with a median CV of 7.96% (Supplementary Data 1—CV_compare with PQ500_peptide; Supplementary Fig. 3a).

PQ500 includes peptides for lower abundant proteins that can usually not be measured in high-throughput neat plasma proteomics but need more sensitive LC-MS methods. Therefore, we also reprocessed a dataset acquired from top 14 protein-depleted samples on a nano-flow LC-MS setup with a 105 min long gradient, which included PQ500 SIL-IS (PXD036594)[36], consisting of depleted plasma samples from 10 controls and 10 Colorectal cancer (CRC) patients. In this dataset, 489 of 804 PQ500 matching endogenous peptides (61% coverage) and 194 of 211 OSPP matching endogenous peptides (92% coverage) were consistently quantified in more than two-thirds of samples. Biological variation increased overall CVs, but endogenous peptides matching the OSPP remained more precise: PQ500 endogenous peptides showed median CVs of 32% (CRC) and 27% (controls), while OSPP endogenous peptides showed median CVs of 31% (CRC) and 25% (controls) (Supplementary Data 1—PXD036594_output; PXD036594_CV_compare with PQ500_peptide). Thus, although PQ500 includes more peptides, the extensive selection of OSPP peptides from large study data had resulted in peptides more consistently detected and also more precisely quantified, even with a more sensitive acquisition method, in both our own, and external datasets.

### Evaluation of the OSPP application in a case study of the human host response to SARS-CoV-2 infection

To demonstrate the usability of our SIL-IS, we applied OSPP as internal standards in a clinical cohort study. We chose a well-characterized clinical cohort of severely ill COVID-19 patients[31,37–39] ($n = 45$, Fig. 3a, Supplementary Data 4—meta data for study cohort). This cohort, enrolled at Charité between March 1–26, 2020, included individuals across the WHO ordinal scale of treatment escalation, which serves as a proxy of disease severity[40]. This reference cohort enrolled patients with WHO levels 0, 3, 4–5, and 6–7, ranging from uninfected to critically ill. Upon proteomic sample preparation using a semi-automated workflow[41], 1 μl (40.4 ng, total peptide amount over all 211 peptides) of the OSPP was spiked into 1.5 μg of total citrate plasma protein digest. The samples were first separated using 20 min, 5 μl/min μflow chromatography on an ACQUITY UPLC M-Class system (Waters), coupled with a ZenoTOF 7600 system (SCIEX). We used a targeted method, Zeno MRM-HR, to compare peptide concentrations across disease states, classified using the WHO ordinal scale for clinical symptom[40] (Supplementary Data 4—WHO score & Severity). Data was processed using Skyline[42], including manual inspection of peak integration.

Peptides detected in more than two-thirds of samples were included for subsequent analysis, and this criterion was met by 202 of the 211 peptides. Among those peptides, when only using the endogenous quantified value, more than half (106/202) were responsive depending on the treatment escalation level (WHO grade). With the application of the OSPP as internal standards, technical bias and noise were better controlled, leading to more precise and reliable results (Supplementary Data 5—Kendall´s Tau trend test_WHO): The improved measurement precision revealed 34 additional peptides with significant changes when quantified with MRM-HR, while 2 peptides that were associated with disease severity without the OSPP normalization lost the significant association—likely these were false positive associations called with the less precise method. As a result, more than two-thirds of peptides quantified with the OSPP (138 out of 202) responded according to COVID-19 severity.

A principal component analysis of peptide quantities (normalized by the OSPP, i.e., endogenous/OSPP, "ratio") revealed the separation of patients according to their treatment escalation levels. The first

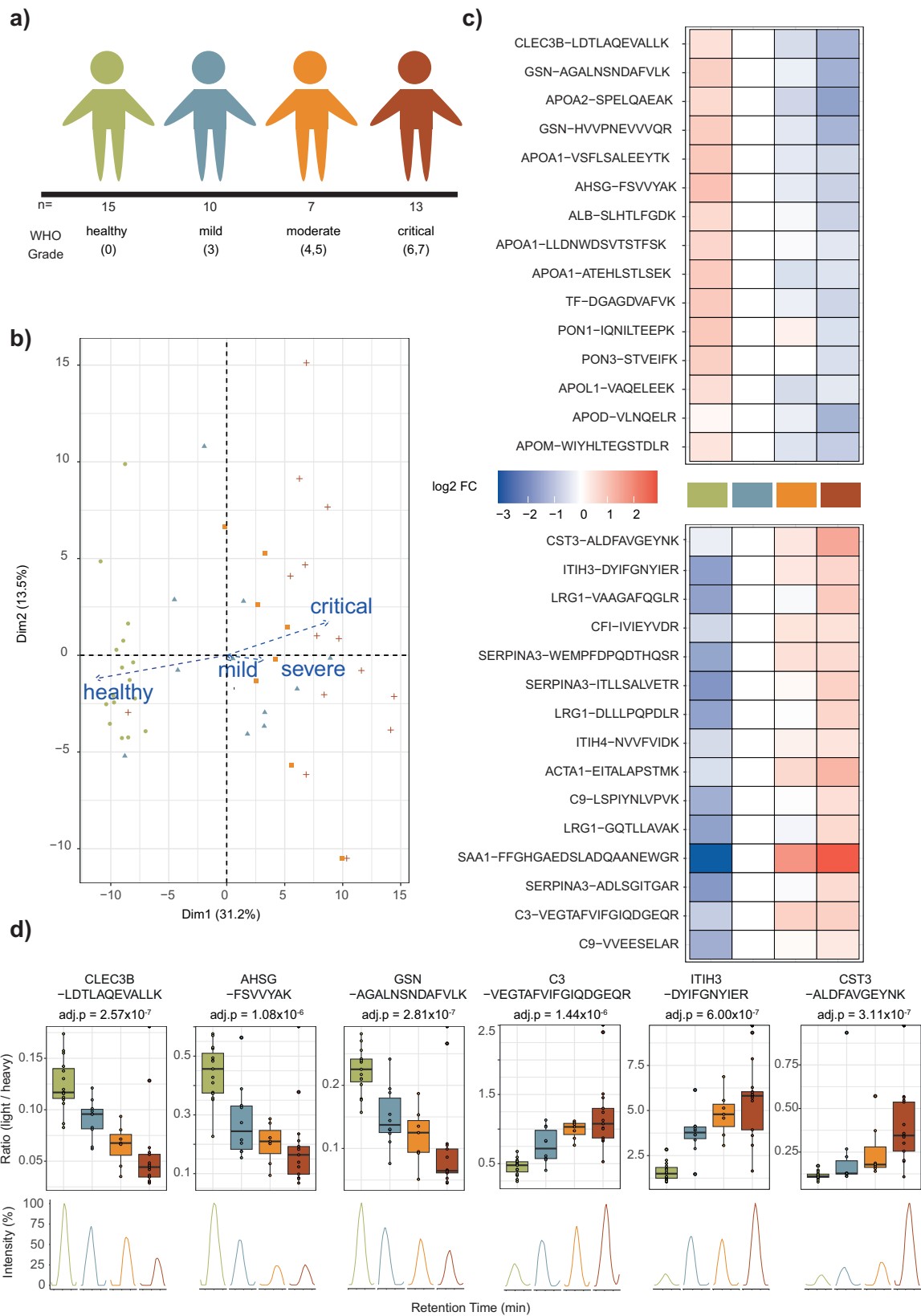

principal component explained 31.2% of the variance and primarily reflected the treatment escalation level (Fig. 3b). A highlight of the top 15 most increased and decreased peptide levels (Fig. 3c) reflects processes important for the host response in COVID-19. For example, peptides derived from tetranectin[43] (CLEC3B) that modulate the biological function and inflammatory process during infection are significantly decreased in severe COVID-19, while complement factor C3-derived peptides[31,38] are induced with severity. Additionally, other peptides exhibited distinct signals corresponding to specific disease escalations. For example, a peptide derived from the enzyme inhibitor ITIH3[44] is strongly associated with infection itself, as is a peptide from the acute phase protein AHSG[39]. Others, such as peptides from the

**Fig. 3 | OSPP peptide quantities from targeted proteomics indicate disease severity in a COVID-19 inpatient cohort. a** Schematic illustration of a balanced cohort consisting of healthy volunteers and hospitalized patients suffering from acute COVID-19 (a subgroup of the PA-COVID19 study[37], the WHO grades are defined in the WHO R&D Blueprint novel Coronavirus[40] and the severity groups are defined based on clinical performance following a recent publication[31]. **b** Principal component analysis (PCA) of OSPP-normalized precursor quantities across samples. Each point represents a sample, colored by disease severity (as indicated in the legend), plotted along the first two principal components (PC1 and PC2) which explain 31.2% and 13.5% of the total variance, respectively. The overlaid arrows represent loadings of selected peptides/proteins, indicating their contribution to the principal components. **c** Peptides with significant abundance change (down- (top panel) and up-regulated (bottom panel)) distinguish healthy from affected individuals, as well as mild from severe forms of the disease, represented by the

WHO treatment escalation score. Heatmaps display the log2 fold-change of the indicated peptide to its median concentration in patients with a severity score of WHO 3. The top 15 significant peptides (adjusted $p$-value < 0.05) are shown. Adjusted p-values were calculated using a p values from two-sided Kendall's Tau trend test based on WHO grade and adjusted for multiple testing using the Benjamini–Hochberg procedure. **d** Visualization of peptide-level responses to COVID-19 severity. Boxplots display OSPP-normalized peptide quantities across patient groups of increasing severity, as described in panel a. The box-and-whisker plots display the 25th, 50th (median), and 75th percentiles in boxes; whiskers display upper/lower limits of data (excluding outliers). Adjusted $p$-values were calculated using a p values from two-sided Kendall's Tau trend test based on COVID-19 severity (healthy, mild, moderate, critical) and adjusted for multiple testing using the Benjamini–Hochberg procedure.

kidney and inflammation marker CST3[31,38], changed drastically during critical COVID-19 cases, indicating the possibility of kidney dysfunction in severe infection[6] and peptides associated with the calcium-regulated, actin-modulating protein gelsolin (GSN[45]) whose reduced abundances were associated with worse outcomes[41] (Fig. 3c, d; Supplementary Data 5−Kendall´s Tau trend test_WHO). Consistent with previous observations, most of the apolipoproteins classically associated with HDLs were less abundant in COVID-19 HDL particles, including APOA-II[46]. In contrast, well-known key acute-phase plasma proteins such as Orosomucoid 1 protein (ORM1)[47] and Serpin A3 Protein (SERPINA3)[48] were increased depend on the severity of COVID-19, in agreement with our previous studies and other reports[39,49,50] (Supplementary Fig. 4).

In summary, peptide relative quantities normalized by OSPP using a targeted proteome method (Zeno MRM-HR) responded according to the treatment escalation levels in a cohort of COVID-19 patients from the early pandemic. Furthermore, this targeted method successfully identified protein markers, covered by OSPP, that were associated with specific disease state transitions.

## Application of the OSPP for cross-method proteome analysis in human plasma

To compare the quantitative performance of OSPP across targeted and untargeted acquisition methods, we measured the same samples in both targeted (MRM-HR) and untargeted (Zeno SWATH DIA[34]) acquisition methods on the same acquisition platform (ZenoTOF 7600) with the same 20 min, 5 μl/min reverse-phase chromatographic gradient. The data processing pipeline involved DIA-NN[28], using the DiOGenes spectral library[29] modified by adding isotopic labels ($^{13}C^{15}N$) specifically on OSPP peptides and excluded b-ions for quantification. The majority of peptides (207/211), both the native and its matched SIL-IS form, were quantified in over two-thirds of patient samples using Zeno SWATH DIA, with 199/211 of these pairs quantified consistently in both Zeno SWATH DIA and Zeno MRM-HR, using the same fragment ions for quantification (Fig. 4a). On the spectral level, the OSPP SIL-IS peptides behaved similarly to endogenous peptides under both Zeno MRM-HR and Zeno SWATH DIA acquisition, with the median fold change of each shared fragment signal from peptide (endogenous / OSPP) being close to 1 (0.9927 for MRM-HR and 0.9938 for Zeno SWATH DIA). This indicated strong spectral agreement and a similar analytical performance between the standard peptides and the respective endogenous peptides. To validate whether OSPP peptides introduced interference in discovery-mode DIA data, we evaluated the distribution of fragment ion intensities within each peptide between acquisition methods. The fold changes of each fragment percentage between MRM-HR and Zeno SWATH DIA were close to 1 (median fold change: 0.9783 for endogenous and 0.9803 for OSPP peptides), demonstrating that both endogenous and OSPP fragment profiles remained consistent across acquisition methods. (Fig. 4b, Supplementary Data 6−Fragment distribution).

To assess the quantitative precision of OSPP on both acquisition platforms, we first generated a study pool sample and analyzed it in quintuplicate, alongside the study samples. As expected, without normalization, the two methods returned different relative values (Supplementary Data 6−foldchange). To correct the different quantification scales between methods, we then applied different normalization strategies. We first tested median normalization based on endogenous peptide quantities, referred to here as "norm_light". After median normalization, the median fold change of all peptide relative quantities between methods increased from 0.006 (before normalization) to 1.18. Next, we used OSPP as the internal standard for normalization. Here, we calculated the ratio of the quantity of the endogenous peptide to that of its corresponding SIL-IS ("ratio"). The relative quantities represented by peptide ratios between methods further improved the 1.06 median fold change of all peptides, indicating improved comparability between methods (Supplementary Data 6−foldchange). The use of OSPP in the DIA method also improved the overall precision. The median CV of all peptide quantities from quintuplicates of the study pool improved from 14% using the median normalization ("norm_light") to 9% of the median CV with the OSPP normalized ratios (Fig. 4c, Supplementary Data 6−CV for studypools (MAD/median)).

We then explored quantitative similarities and differences of all samples between the targeted Zeno MRM-HR and the DIA method by fitting a linear regression model to each peptide quantity in each severity group. In our comparison, an $r^2$ close to 1 indicates data acquired from Zeno SWATH DIA has a close relation to Zeno MRM-HR data, and a slope close to 1 indicates that the quantities obtained from both methods are highly consistent. For example, the quantity for the Apolipoprotein A-II (APOA2)-derived peptide SPELQAEAK correlated with an $r^2$ of 0.49 using median normalization; the correlation was improved to 0.98 using OSPP normalized ratios (Fig. 4d, Supplementary Data 6−R2-slope). Across the top 30 up- or down-regulated peptides that distinguish the COVID-19 patients in Zeno MRM-HR (as in Fig. 3c), the average $r^2$ of the mild patient group was 0.53 (peptides that increased in concentration in more ill patients) or 0.27 (decreased) without the standard and improved to an average $r^2$ of 0.88 (increased) or 0.74 (decreased) upon OSPP normalization. Also, across all peptides in all severity groups, the median correlation between the normalized quantities obtained with Zeno MRM-HR and Zeno SWATH DIA improved markedly from 0.39 (normal_light) to 0.83 (ratio). For quantity comparison between methods, we also observed the median slope being generally closer to 1 (0.91) when applying normalization to the OSPP internal standard (Fig. 4f, Supplementary Data 6−R2-slope).

Due to the higher technical precision upon normalization to the internal standard, more proteins were assigned to disease severity. For example, Apolipoprotein A-II (APOA2), a key component of high-density lipoprotein (HDL), has been reported to be dysregulated in patients with COVID-19, reflecting alterations in lipid metabolism and the host inflammatory response[51–53]. In our analysis, with normalization

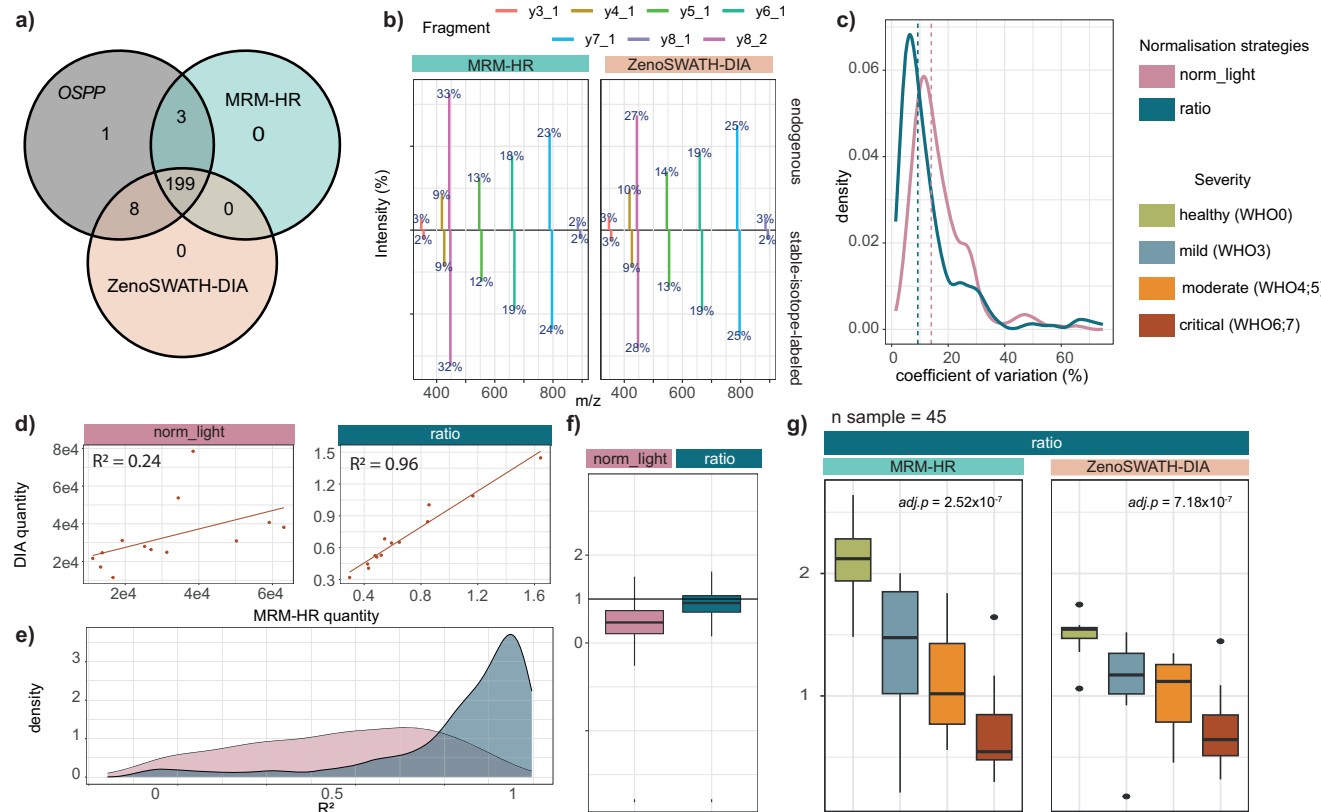

**Fig. 4 | Alignment of targeted and untargeted MS data by using the OSPP as an internal standard, exemplified on a COVID-19 cohort (Source data are provided as a Source Data file). a** Overlap of quantified peptides from targeted and untargeted MS methods to the full OSPP peptide list. **b** Average fragment ion spectral distribution SPELQAEAK from Apolipoprotein A-II in triplicate study pool injections in both MRM-HR and Zeno SWATH DIA data. The percentages represent the median intensity percentage of each fragment ion from the same precursor ion (with a maximum of 12 fragments being included, summing to 100%). The upper panel shows endogenous fragment ion distribution, and the lower panel shows the distribution of fragment ions of OSPP peptides. **c** Coefficient of variation (CV) values were calculated from each peptide quantity in quintuplicates of a study pool sample for both methods (density plot). Dashed lines display the median CVs of each normalization strategy. **d** Quantities of SPELQAEAK in critically ill COVID-19 patients ($n = 13$), as determined by both MS methods (correlation plot). A reference line (y - x) was added for comparison. **e** Distribution of all $r^2$ calculated from

correlating normalized peptide quantities from Zeno MRM-HR and Zeno SWATH DIA (median normalized endogenous quantities in purple; ratio in blue) in each severity group. **f** Distribution of slopes from correlating peptide quantities derived from Zeno MRM-HR and Zeno SWATH DIA methods, fitted to a linear model (y - x) (Boxplot). The box-and-whisker plots display the 25th, 50th (median), and 75th percentiles in boxes; whiskers display upper/lower limits of data (excluding outliers). A vertical line at 1 showing how close the median slope is to 1. **g** Median endogenous normalized ("norm_light") and OSPP normalized ("ratio") quantity of Apolipoprotein A-II-derived peptide SPELQAEAK across patient groups of increasing severity (Boxplots). The box-and-whisker plots display the 25th, 50th (median), and 75th percentiles in boxes; whiskers display upper/lower limits of data (excluding outliers). Adjusted p-values were calculated using a $p$ values from two-sided Kendall's Tau trend test based on COVID-19 severity (healthy, mild, moderate, critical) and adjusted for multiple testing using the Benjamini–Hochberg procedure.

to the internal standard, a significant association of APOA2-derived peptide SPELQAEAK with both targeted and untargeted proteomic methods was detected (Fig. 4g). Overall, 132/199 peptides quantified by MRM-HR and 137/199 peptides from various OSPP concentration bins quantified by Zeno SWATH DIA show significant association with the WHO score (Supplementary Fig. 5, Supplementary Data 6–Kendall´s Tau trend test_Severity).

## Use of the OSPP in comparing plasma proteomic data acquired on different DIA-MS platforms

One of the key motivations of including a SIL-IS is to achieve cross-platform comparability of the acquired data, which remains a key challenge in clinical proteomics. We thus expanded the evaluation to various LC-MS configurations, spanning from nano-flow chromatography (250 nl/min) to an 800 µl/min analytical flow rate chromatography used in high-throughput applications. These LC systems were coupled to different mass spectrometers, namely an Orbitrap Exploris 480 System (Thermo Scientific) and a ZenoTOF 7600 System (SCIEX). We injected samples from the aforementioned COVID-19 cohort, and

data was acquired using DIA-MS on all platforms and processed with the afore-described DIA-NN-based pipeline.

Out of the 211 internal standard peptides of the OSPP, 187 pairs (88.6%) of the endogenous and its matched SIL-IS peptides were consistently identified and quantified across more than two-thirds of the samples, across three LC-MS platforms and five different acquisition and quantification methods (Fig. 5a, Supplementary Data 7–Detection of peptide in certain Severity Group). Retention times (RTs) were consistent between OSPP and respective endogenous peptides, with a median difference less than 200 ms, indicating that the peptides were correctly identified (Supplementary Data 7–Retention time of each sample). Moreover, the RTs for each peptide remained stable across all samples, with RT fluctuations across all samples within the acquisition batch below 40 seconds and a median CV less than 2% in each platform (Supplementary Data 7–RT difference between light & OSPP). Also, fragment ion intensity distributions were comparable between endogenous peptides and their labeled counterparts (Fig. 5b, Supplementary Data 7–Fragment distribution). We performed inter-class coefficient correlation (ICC) of all samples across platforms. More

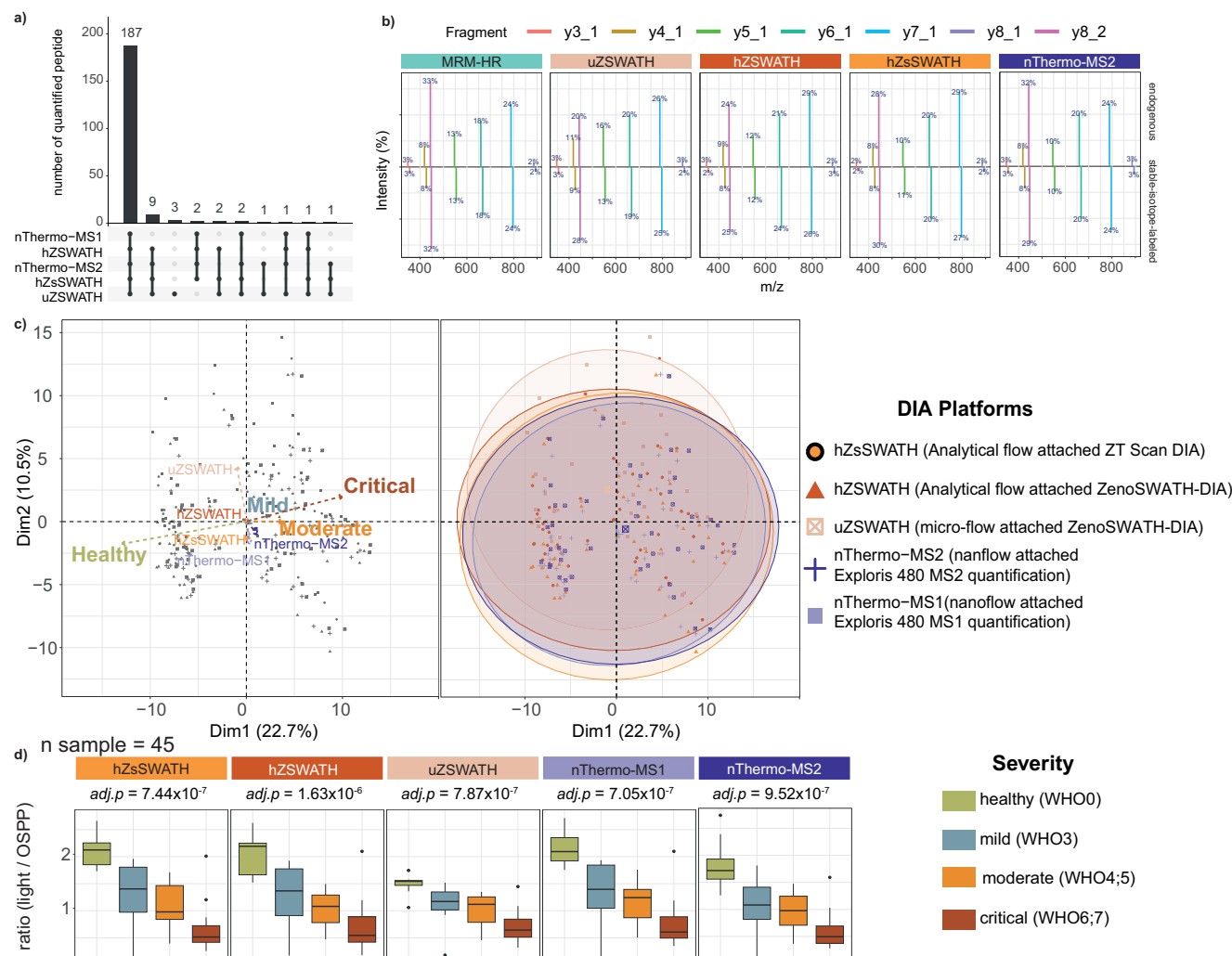

**Fig. 5 | Data alignment across DIA-MS platforms using the OSPP as an internal standard, exemplified on a COVID-19 cohort (Source data are provided as a Source Data file). a** Intersections of quantified OSPP peptides per DIA-MS platform Each column corresponds to the number of quantified peptides per DIA-MS platform or sets of platforms (highlighted by dots connected with vertical lines). The number of peptides per set appears above the column. The names of each DIA-MS platform are indicated on the left (abbreviations see color legend). **b** Average fragment ion spectral distribution of Apolipoprotein A-II-derived peptide SPELQAEAK in triplicate study pool injection in each DIA acquisition method mentioned in Fig. 5a. The percentages represent the median intensity percentage of each fragment ion from the same precursor ion (with a maximum of 12 fragments being included, summing to 100%). The upper panel shows endogenous fragment ion distribution, and the lower panel shows the distribution of fragment ions of OSPP peptides. **c** Principal component analysis (PCA) of 187 OSPP-normalized precursor quantities across samples, visualizing variation in patient proteomes by COVID-19

disease severity. Each point represents a sample, colored by disease severity (as indicated in the legend), plotted along the first two principal components (PC1 and PC2), which explain 22.7% and 10.5% of the total variance, respectively. In the biplot (left panel), vectors indicate loadings for variance, with the direction and length of the vectors reflecting their contribution to the principal components. The right panel displays the same scores; each point represents an individual sample, shaped and colored according to the acquisition platform, overlaid with confidence ellipses (95%) for each platform, illustrating sample dispersion and overlap. The almost congruent ellipses indicate uniform variance within groups. **d** OSPP normalized quantities of Apolipoprotein A-II-derived peptide SPELQAEAK across different severity groups. The box-and-whisker plots display the 25th, 50th (median), and 75th percentiles in boxes; whiskers display upper/lower limits of data (excluding outliers). Adjusted p-values were calculated using a p values from two-sided Kendall's Tau trend test based on COVID-19 severity (healthy, mild, moderate, critical) and adjusted for multiple testing using the Benjamini–Hochberg procedure.

than 90% of the fragments showed ICC > 0.75, and 85% exceeded 0.9 (median ICC of 0.967 for endogenous peptides and 0.987 for spiked OSPP) (Supplementary Data 7−ICC fragment percentage of all peptides). Furthermore, peptide-level analysis revealed that correlations between peptides mapping to the same protein showed diverging overall correlations; in each platform, over 40% of same-protein peptide pairs had an $R^2$ above 0.7 (Supplementary Data 7−correlation between peptides from same protein). It is worth speculating that the high isoform diversity in the human plasma is a main contributor to the diverging quantities of peptides derived from the same protein model.

To check the quantification consistency across platforms, we compared the peptide quantification across acquisition platforms for

all COVID-19 clinical samples. Due to the different design of the instruments, as expected, peak areas, which represent relative quantities, differed between platforms without normalization to the internal standard. Upon forming ratios with the corresponding OSPP standard, the peptides were quantified across platforms with a median CV of just 13.2% (Supplementary Data 7−CV for studypools(MAD/median)).

In a principal components analysis (PCA) of OSPP-normalized ratios for 187 peptides quantified across all platforms, the data showed strong cross-platform consistency. The first principal component (PC1, 22.7%) corresponded to the major variation associated with disease severities, while PC2 (10.5%) was mostly driven by technical variance

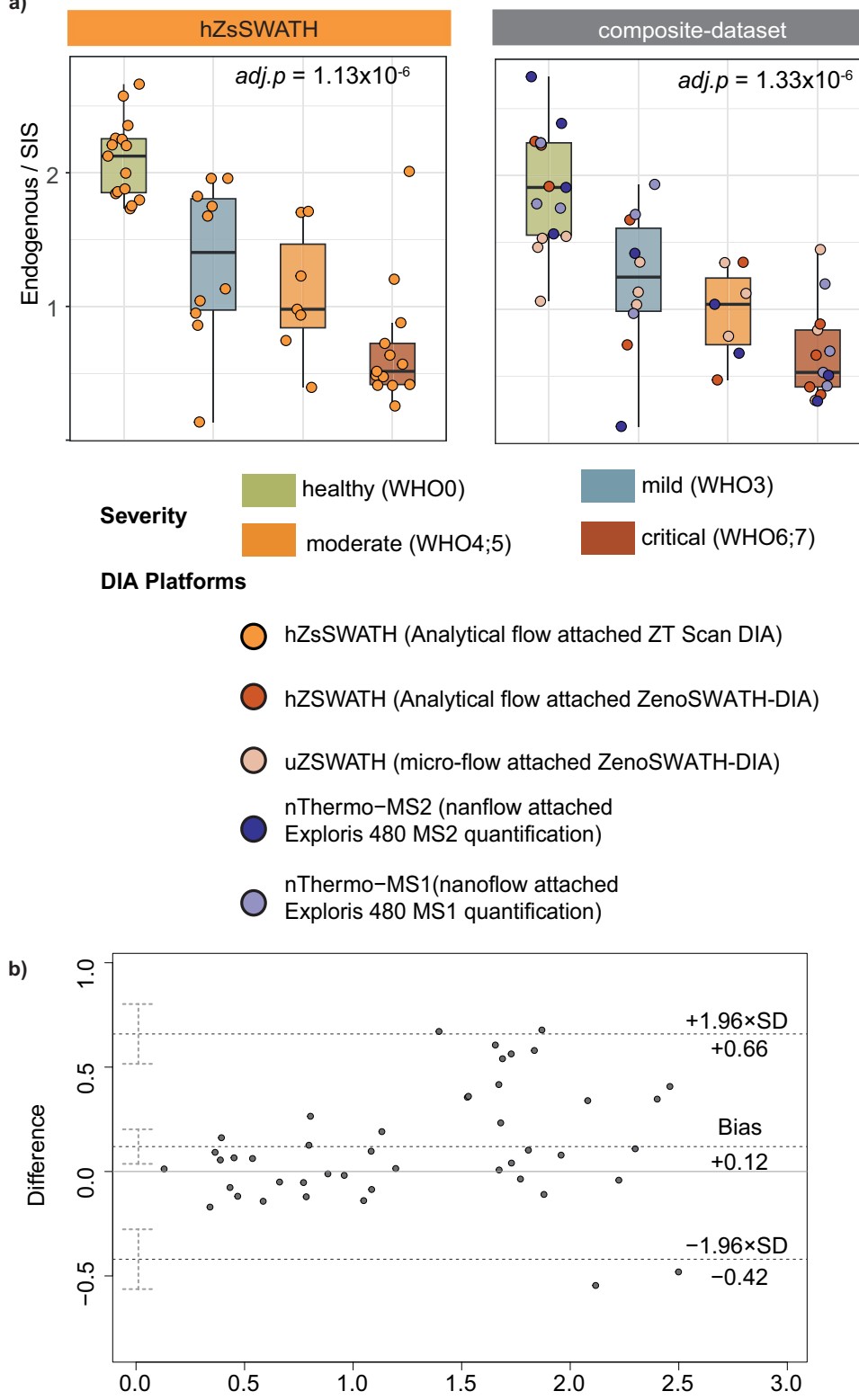

**Fig. 6 | Data alignment across single-platform and composite dataset using the OSPP as an internal standard, exemplified on a COVID-19 cohort (Source data are provided as a Source Data file). a** Normalized quantitative ratios of Apolipoprotein A-II-derived peptide SPELQAEAK from the analytical flow-rate ZT Scan DIA (hZsSWATH) method and a "composite dataset" containing samples with the same severity distribution and from various DIA acquisition platforms (other than hZsSWATH). The box-and-whisker plots display the 25th, 50th (median), and 75th percentiles in boxes; whiskers display upper/lower limits of data (excluding outliers). Adjusted *p*-values were calculated using a *p* values from two-sided Kendall's Tau trend test based on COVID-19 severity (healthy, mild, moderate, critical) and adjusted for multiple testing using the Benjamini–Hochberg procedure. **b** Quantitative comparison of the analytical flow-rate ZT Scan DIA (hZsSWATH) dataset and "composite dataset" via a Bland–Altman plot. Shown are the mean difference and 95% confidence limits for peptide quantities (ratios) of Apolipoprotein A-II-derived peptide SPELQAEAK between all samples measured with analytical flow-rate ZT Scan DIA compared with the "composite dataset" that constitutes samples from various DIA acquisition platforms (Supplementary Data 7–Bland-Altman output).

between platforms (Fig. 5c, left panel). Visualization using 95% confidence ellipses highlighted the substantial overlap of data points across platforms, indicating minimal technical bias and consistent peptide quantification between different LC-MS platforms after OSPP normalization (Fig. 5c, right panel). Variance partitioning analysis across peptides further confirmed that OSPP-based normalization substantially reduced technical variance attributs to cross-platform differences, from a median of 39.6% among all peptide quantities using median normalization down to 4.8% after OSPP normalization. (Supplementary Data 7−variance partitioning across peptides). All above analysis demonstrates that OSPP normalization effectively minimizes cross-platform technical variance and enables consistent quantification across DIA platforms. For instance, the OSPP normalized ratio of peptide SPELQAEAK (from Apolipoprotein A-II) showed consistent quantification across all DIA platforms, and this normalized ratio effectively allows comparable relative quantities and helps distinguish COVID-19 severity across platforms (Fig. 5d).

To further highlight the possibility for cross-platform data alignment using the OSPP internal standard, we evaluated whether data collected with different acquisition platforms can be mixed to produce comparable results to those from a single platform. We created an artificial "composite dataset" with peptide quantities randomly taken from datasets acquired from various platforms (OSPP normalized), reflecting the sample and severity distribution of the original clinical cohort (Supplementary Data 4−Composite dataset). To verify whether the artificial dataset showed similar peptide expression changes across severity levels, we compared it with data acquired using analytical flow on the ZT Scan DIA platform. Among the 159 peptides that were above the lower limit of quantification (LLOQ), 54 peptides showed substantial abundance changes in both the composite dataset and the single-platform cohort when normalized using the OSPP. Additionally, 10 peptides showed significant changes across severity groups ($p.adjust < 0.05$) only in the single-platform cohort, likely due to platform-specific sensitivity and outlier effects near the detection limit. Conversely, while 11 peptides did not show statistically significant changes across severity groups in the single-platform dataset, they displayed significant differences in the composite dataset. In general, normalization to the OSPP internal standard allowed cross-platform combination of the proteomic results. For example, the concentration of Apolipoprotein A-II (represented by peptide SPELQAEAK) is significantly increased during COVID-19 progression, as observed on the ZT Scan DIA platform, and this trend is similarly confirmed in the composite dataset (Fig. 6a, Supplementary Data 7−Statistics_composite dataset).

To assess quantification consistency between the composite dataset and individual dataset, we performed Bland−Altman analysis for each peptide, using all corresponding sample measurements from both datasets (Supplementary Data 7−Bland-Altman output). To illustrate a representative example, we choose the APOA2-derived peptide. Using the OSPP normalized ratio, we observed strong agreement on quantitative data obtained with both cohorts (Bias = 0.12), with only a few data points falling outside the limits of agreement (Fig. 6b). Bland−Altman analysis across all peptides that were above the limit of quantification showed a median absolute bias of 0.037 between datasets, with an average limits-of-agreement range of 1.716 units. Notably, 76.1% of peptides demonstrated differences within ±10%, indicating high concordance between quantification methods and 96.5% of the peptides showing bias between −1 and 1, signifying significant concordance in peptide ratios between the two datasets (Supplementary Data 7−bland-altman output); Although a small subset of peptides ($n = 6$) showed clear bias (absolute bias > 1), these likely reflect a minority influenced by intrinsic technical or biological variability. The overall near-zero median bias supports robust quantification across the majority of peptides between platforms, affirming the reliability of the method despite inherent complexities.

## Discussion

With the growing focus on plasma and serum proteomics, there is an increasing need for workflows that enable cross-platform and cross-study comparability. While absolute protein quantification remains the ultimate goal, absolute quantification at the peptide level is easier to implement in routine workflows and represents a significant step towards improving analytical precision and accuracy and simplifies platform alignments[16,20−22,26,27,31]. In this study, we introduced the Charité Open Standard for Plasma Proteomics (OSPP), an effective, disease-agnostic, and extendable peptide internal standard designed for neat plasma and serum proteomic studies. In contrast to earlier or disease-specific SIL-IS or marker panels[31,54−56], OSPP consisted of peptides selected on the basis of their analytical performance in 1,505 repeated runs spanning over 15,000 human samples across eight large studies, all analyzed on the same proteomic platform (Fig. 1). In the selection procedure, we prioritize consistent detectability, low variance, and efficient synthesis. After combining the successfully validated peptides in a SIL internal standard panel, we validated its performance across sample matrices, acquisition methods, and instruments. The resulting internal standard features evenly distributed peptides across the chromatographic gradients, minimal retention-time variance, and consistent quantification across several tested proteomic setups.

A major challenge in LC-MS-based proteomics is the alignment of data acquired across different LC-MS setups, arising from variability in chromatographic separation, instrument design, ionization efficiency, or acquisition methods[57,58]. The use of a spiked-in internal standard allows addressing these limitations. Indeed, our cross-platform evaluations were representative of the situation and confirmed that different LC−MS platforms, between targeted and discovery acquisition methods, produce raw quantification outputs on vastly different scales, in some cases differing by several orders of magnitude for the same peptide. We performed cross-platform evaluations on a set of clinical samples spanning multiple LC−MS instruments and acquisition methods and found that with the use of the SIL-IS, comparable quantification performance between platforms is achieved. Moreover, when benchmarking OSPP across samples from different biological matrices, we report matrix dependency in the peptide ranks; however, the majority of OSPP peptides remained consistently identified. Finally, also within a sample matrix, the use of the OSPP panel was beneficial. For example, due to the increased technical precision, additional proteins could be associated with disease severity in a COVID-19 cohort, while two protein associations were lost. These were most likely false-positive associations in the noisier dataset analyzed without the standard and might related to the cohort used. Notably, some proteins, such as C3 or CST3 are increasing in the presented cohort but not across all COVID-19 studies[59,60]. These differences likely reflect cohort-specific effects.

Despite being selected for purely technical reasons, many of the 211 OSPP peptides are derived from centrally important plasma proteins, which function in metabolism and nutrient transport, the innate and adaptive immune system, and the coagulation system (Fig. 2). As these proteins respond in a number of diseases and encompass many proteins that were already established clinical biomarkers and/or FDA-approved drug targets, the OSPP can also monitor many disease processes directly, e.g., without being expanded by disease-specific markers. For example, we applied the OSPP in a well-characterized COVID-19 cohort[31,37−39], where we detected many disease-responsive proteins. In a parallel study, we applied the OSPP in the characterization of Loiasis, a rare and neglected tropical disease caused by the African eye worm *Loa Loa*. OSPP facilitated identification of potential biomarkers and classifiers of disease stage[61].

As a disease-agnostic internal standard, OSPP is not designed as a clinical marker panel but to adapt and improve data quality in diverse blood-based proteomics studies. Its open design allows users to use it

for different blood-based studies[61] and to add further disease-specific peptides. In theory, OSPP can also be integrated with other SIL-IS panels, such as PQ500[23], for a broader analysis. Thus, by adding additional peptides or by recombining individual peptides into new panels, the OSPP can be converted into both smaller disease- or indication-specific marker panels or expanded to capture a broader range of the proteome. Moreover, works in metabolomics and microbiome proteomics have shown that SIS are not limited to improving the quantification, or to estimating absolute concentrations, of the directly matched, endogenous molecules but can potentially also be used as surrogate internal standards for a much broader set of peptides and proteins[62–64]. We anticipate that future algorithmic developments could unlock this potential also for proteomics, opening new opportunities for broader quantitative harmonization.

Despite the above strengths of the OSPP, limitations in the use of a SIL-IS should be acknowledged. For example, all data used for peptide selection were acquired using the same mass spectrometry platform, using Scanning SWATH acquisition methods[38]. Although we tested the final peptide standards and obtained excellent analytical performance on a series of different instruments and acquisition methods, this selection constraint means that peptides that are not well detected on the selection platform are lacking in the OSPP standards.

A future challenge involves enabling hybrid analyses that simultaneously perform absolute quantification using internal standards and discovery-based proteomics within a single DIA experiment. Although current software tools such as DIA-NN[28] or Spectronaut[65] partially support this, these functionalities could still be optimized, as they were not originally designed specifically for this purpose. We provide a protocol for generating an OSPP-inclusive spectral library along with a corresponding data processing pipeline optimized for DIA-NN. While the inclusion of SIL-IS peptides increases data processing complexity, many bioinformatic processing steps that are required further downstream profit from the presence of an internal standard and can be simplified. For example, because SIL-IS peptides provide internal correction for each sample, intensity distributions do not require alignment, and biases such as instrument performance variation and time-dependent drift are inherently corrected. This situation greatly simplifies the integration and joint analysis of samples from different batches, instruments, or sites.

In this study, OSPP as an internal standard supports quantification using a simple single-point calibration strategy, forming direct ratios between endogenous peptides and their isotopically labeled standards. This simple approach effectively minimized technical variance and proved suited for a range of studies, prioritizing consistent quantification with minimal measurement and data analysis time. Indeed, this strategy also mitigated batch effects, which, in label-free proteomics, can require significant hands-on bioinformatics time. However, single-point calibration has limitations, particularly when large concentration differences exist or when flexibility in standard composition is needed. In such cases, matrix-matched standard dilutions provide greater accuracy[66].

While OSPP-based internal standard workflows provide robust quantification at the peptide level, inferring accurate protein-level abundances remains a broader challenge in bottom-up proteomics, and specifically in plasma, which is rich in protein isoforms. Thus, while peptide-centric quantification remains robust and reproducible, reliable protein-level estimation, especially in complex matrices like plasma, remains non-trivial. Computational methods such as MaxLFQ[67] have improved protein inference from peptide intensities, yet the complexity of the proteoform space creates limitations, not the least of which is that some information is inadvertently lost by digesting the intact proteins into peptides. In this study, we chose not to further address the pre-analytical factors, such as blood handling and processing, which remain a key challenge in plasma proteomics. Nonetheless, the use of an internal standard like OSPP can help disentangle technical from pre-analytical variation by providing stable reference signals across samples. This capability is important to extend OSPP's utility as an internal standard for reliable application in diverse clinical proteomics contexts and multi-laboratory clinical studies.

In summary, with the Charité Open Peptide Standard for Plasma Proteomics (OSPP), we present a robust and openly accessible peptide internal standard to improve reproducibility and cross-platform quantification alignment of plasma and serum proteome studies. By selecting peptides from large proteomic datasets to identify peptides with ideal properties, the OSPP peptides are concentration matched to endogenous peptides and consistently detectable across acquisition methods and platforms. The open design of our standard enables researchers to order part or all peptides from any peptide biosynthesis providers, allowing the creation of highly cost-effective internal standards in-house. In this way, the OSPP is resilient, versatile, and cost-effective. We demonstrate the application of the OSPP on the use case of a COVID-19 cohort study to achieve consistent proteome data across blood matrices, platforms, and acquisition methods, using both targeted, discovery, and hybrid workflows. Thus, OSPP not only allows customization for disease-specific marker panels but also provides a robust, reproducible platform that captures key plasma proteins and FDA-approved drug panels with proven biomedical utility themselves, demonstrating its utility for quantitative proteomics and patient stratification in severe COVID-19.

## Methods

### Reagents

Water was obtained from Merck (LiChrosolv LC-MS grade; Cat# 115333), acetonitrile from Biosolve (LC-MS grade; Cat# 012078), trypsin (Sequence grade; Cat# V511X) from Promega, 1,4-Dithiothreitol (DTT; Cat#6908.2) from Carl-Roth, iodoacetamide (IAA; Bioultra; Cat# I1149) and urea (puriss. P.a., reag. Ph. Eur.; Cat#33247) from Sigma-Aldrich, ammonium bicarbonate (Eluent additive for LC-MS; Cat# 40867) and Dimethyl sulfoxide (DMSO; Cat# 41648) from Fluka, formic acid (LC-MS Grade; Eluent additive for LC-MS; Cat# 85178) from Thermo Scientific™, bovine serum albumin (BSA; Albumin Bovine Fraction V, Very Low Endotoxin, Fatty Acid-free; Cat# 47299) from Serva., commercial human plasma samples (Human Source Plasma, LOT# 20CILP1034) from zenbio.

### Peptide selection and synthesis

To prioritize the most reliably quantified precursors and minimize the influence of such factors as precursor abundance, study cohort, MS setups, LC separations, and sample preparation procedures, we first selected a wide range of studies with different sample types (Supplementary Data 1–selection cohort). These samples were study pools prepared from human plasma and serum using a semi-automated workflow in 96-well plates, which involves a cleanup step using solid-phase extraction before being analyzed on a proteomic platform that uses analytical flow rate reverse-phase chromatography with water-to-acetonitrile gradients and a throughput of 5–8 min/sample[38]. Proteomes were recorded using data-independent acquisition on two TripleTOF 6600+ instruments (Sciex) operating in SWATH[68] or Scanning SWATH[38] mode. Data was analyzed using DIA-NN[28] with the DiOGenes spectral library[29].

Then we introduced a relative rank metric, which was defined as follows. First, we defined precursor weight as a ratio of a precursor's % presence $PPres$, to the coefficient of variation (CV), see Eq. (1), and a weight-based rank, see Eq. (2). Here, $p$ stands for precursor and $n$ for a study pool series. The weight thus corresponds to a precursor's signal-to-noise ratio ($S/N = 1/CV$) multiplied by its presence and reflects our desire to pick up precursors that are reliably detected (showing high presence) and reliably quantified (showing high $S/N$). The measurements are conducted on an intensity scale; therefore, to get an adequate evaluation of variation in the data, we used CV calculated on the

intensity scale as the ratio of intensity standard deviation $\sigma_s(I(p,s,n))$ to its mean $mean_s I(p,s,n)$:

$$CV(p,n) = \sigma_s(I(p,s,n))/mean_s I(p,s,n) \qquad (7)$$

The number of precursors was changing from study to study, with min = 4653 and max = 6361 (the full peptidome consisted of 10560 precursors, of which 7621 were proteotypic). To minimize the influence of the total number of precursors on the ranking, we introduced relative rank $RelRank(p,n)$, defined as the ratio of the precursor's rank $Rank(p,n)$ to the maximum rank value $max_p\{Rank(p,n)\}$ in a study, see Eq. (3).

Finally, the precursor's average (over considered studies) relative rank $RelRank(p)$, see Eq. (4), was used to select the best „global" (i.e. non-project-specific) precursors for every protein, while we also required that the lower cutoff of the relative rank be set as 0.5, see Eq. (5). This threshold guarantees that selected precursors on average have CV below the median CV.

Additionally, we only consider proteotypic peptides in our panel and for more reliable quantification, require those peptides to be quantified in at least half of the projects ($\geq$ 4 projects) and to be present in at least 2 blood matrices, see Eq. (6). This threshold (half of the projects) is a compromise between sensitivity and specificity— with presence in a lesser number of projects, we could measure more study-specific proteins, but at the cost of not measuring them in some other projects. This requirement was complemented by the requirement for the presence of the peptide in at least two sample matrix types. To avoid all peptides coming from those top abundant proteins in plasma and to allow covering a larger dynamic concentration range of proteins, no more than top 3 peptides are selected for each protein. Eventually, this selection process identified 382 consistently quantified peptides, approximately 5% of considered proteotypic precursors. Of them, 161 (42%) were detected in all eight studies with an average presence of 97% and a relative CV of 0.5. For 49 peptides detected in only four studies average peptide presence was 95%, and average RelCV ~ 0.5. All other peptides, detected in 5–7 studies, had an average presence of 96% and an average RelCV ~ 0.5.

## Further selection based on physical-chemical-and analytical properties

The chemical properties of each peptide were calculated using the R package "Peptides v2.4.6". The hydrophobicity of each peptide was calculated using the function "hydrophobicity_kyte"[69], the hydrophobicity scales run from −2 to 2, where 94 peptides are hydrophobic (>0) and 117 are hydrophilic (<0); net charge is calculated with function "charge"; high missed cleavage is considered and excluded when "KK | KR | RR | RK | KP | RP" appears in the peptide sequence, with the exception of peptide "ANRPFLVFIR" (SERPINC1) which we previously found to be of interest and with good performance across large numbers of samples[31]. Peptides containing cysteine and N terminal glutamine that are easily modified are excluded, except "IC(Carboxymethylated)LDLQAPLYK" which is the only selected peptide for protein "PF4". An additional 24 peptides (30 peptides, 6 of which are also selected from previously mentioned study pools selection) from the previous MRM panel[31] were included in the list. For checking likelihood of successful synthesis of peptides, the Peptide Synthesis and Proteotypic Peptide Analyzing software tool (Thermo, [34]) was used with synthesis.

A pool of all the study pools used for the initial selection was prepared and analyzed on a 20 min water-to-acetonitrile 5 μl/min microflow-rate chromatographic gradient analyzed by high-resolution multiple reaction monitoring (Zeno MRM-HR) on a ZenoTOF 7600 instrument (SCIEX) to check the analytical performance of all short-listed peptides. Nearly all peptides were well identified with a charge state, mostly 2 or 3. One peptide, EGPYSISVLYGDEEVPRSPFK, from protein FLNA failed to be identified on μflow; however, it has a good identification rate on the analytical flow LC-attached MS instrument with a charge state of 4.

All the above criteria are listed in Supplementary Data 1—peptide properties.

## Peptide synthesis and validation

Reference peptide standards were synthesized by Pepmic Co., Ltd (Suzhou, China) Native peptides (natural, light [NAT]) were obtained at ≥95% purity and stable isotope-labeled heavy labeled peptides (labeled on C-terminal lysine (K) or arginine (R) with stable isotopes (K(U-$^{13}$C$_6$,$^{15}$N$_2$) or R(U-$^{13}$C$_6$,$^{15}$N$_4$)))—at ≥70% purity. Validation of the synthesized peptides involved initial assessment via LC-UV/VIS and LC-MS analysis.

All peptide stock solutions were prepared at 1 mg/ml in 50:50 (v/v) ddH2O: acetonitrile mix. The peptides were pooled in groups of 11 (~ 20 peptides per group) of each native and isotopically labeled standard, based on their endogenous abundance in the EDTA plasma pool of all the study pools acquired by μ-flow DIA MS. The peptide pools were further analyzed using the same LC-MS method. The validation of peptide synthesis was considered successful after passing following two criteria: all isotopically labeled peptides should coelute with their corresponding native forms in chromatograms, and no native peptide was identified in isotopically labeled-only pools, confirming the satisfactory purity of approximately 70% and affirming their successful synthesis and compatibility with our analytical platform. All synthesis peptide standards passed the above criteria and are aliquoted and stored in 96-well plates in a -80 °C freezer for future preparation.

## Generation of the OSPP mixture

We first mixed all isotopically labeled heavy peptide standards to reach a final concentration of 1 μg/μl and conducted dilution series with 1/10/100/300/900 pg/μl of each peptide. The signal ratio (native endogenous peptide signal / heavy isotope labeled peptide signal) is calculated for each peptide in each concentration. For selecting an appropriate concentration of each peptide, we first calculate the linearity of each peptide within 1–900 pg/μl. Among the linear concentrations, only the concentrations where heavy peptide quantities closely match their native counterparts within a 2x log10 difference were chosen. The concentration of each peptide was further adjusted and calculated to make sure all heavy peptides' signals were the same or at most within a log10 difference from their endogenous counterparts. Next, we categorized all peptides into four distinct concentration tiers, mixing to establish a comprehensive concentration range of 10 pg/μl to 2 ng/μl of each peptide within the OSPP mixture (Supplementary Data 2). To avoid possible evaporation, the OSPP are diluted in 10% v/v acetonitrile, exhibiting no discernible evaporation effects when mixed with digested plasma samples in 384-well plates. We tested the performance of the OSPP by spiking 1 μl (40.4 ng for all 211 peptides) into every 1.5 μg of digested plasma pool; signals of all peptides fell within log10 difference to their respective endogenous signals.

## Equally-concentrated ("Single-conc. Std")

"Single-conc. Std" was prepared by pooling the same amount of each peptide. In the mixture, all peptides are equally concentrated with 600 pg/μl of each and housed in 50% acetonitrile. For matrix performance tests, the single-conc. Std was in 100 pg/μl as diluted in 10% acetonitrile.

## Clinical study design and participants

Patient samples were collected as part of an observational cohort[38,39,41]. The study protocol, including patient characteristics, treatment, and

clinical outcomes, has been described previously[37,70]. Briefly, all hospitalized patients with PCR-confirmed SARS-CoV-2 infection treated at Charité−Universitätsmedizin Berlin, a tertiary care center, were eligible for inclusion regardless of age, sex, or disease severity, following written informed consent. To reduce selection bias toward less severe cases, patients requiring invasive ventilation were also included under a deferred consent procedure. The study size was determined by patient availability and logistical feasibility, and all enrolled patients were included in the analyses. Study date cutoffs were the 1st to 26th of March 2020. The study is registered in the German and the WHO international registry for clinical studies (DRKS00021688). The study was approved by the ethics committee of Charité´−Universitätsmedizin Berlin (EA2/066/20). The cohorts are summarized in Supplementary Data 4−meta data for study cohort.

## Ethics

The COVID-19 cohort is a subcohort of the Pa-COVID-19 study conducted at Charité−Universitätsmedizin Berlin, Germany[37,71], and the matrix test cohort is part of prospective observational cohort study. Both studies were carried out in accordance with the Declaration of Helsinki and the principles of Good Clinical Practice (ICH 1996), where applicable, and were approved by the ethics committee of Charité−Universitätsmedizin Berlin (EA2/066/20, EA4/245/20). Written informed consent was obtained from all participants or their legal guardians before initiation of study procedures. All data have been de-identified and presented in aggregate, with age reported as ranges to prevent identification, while maintaining sex/gender reporting in accordance with journal policy.

## Blood sample acquisition

Blood collection tubes (EDTA, citrate, heparin, and serum, respectively) were centrifuged at 2000G, 4 °C for 15 mins, and plasma or serum, respectively, was collected and frozen at -80 °C for future analysis.

## Sample preparation

**Plasma samples and BSA.** Samples were prepared with minor modifications as described previously[41]. Briefly, plasma/serum samples were stored at −80 °C for 24–36 months prior to preparation, and clinical samples and calibration series were prepared as follows: 5 µl of citrate plasma were added to 55 µl of denaturation buffer, composed of 50 µl 8 M Urea, 100 mM ammonium bicarbonate, 5 µl 50 mM dithiothreitol (DTT) and an internal standard mix. The samples were incubated for 1 h at room temperature (RT) before the addition of 5 µl of 100 mM iodoacetamide (IAA). After a 30 min incubation at RT, the samples were diluted with 340 µl of 100 mM ammonium bicarbonate and digested overnight with 22.5 µl of 0.1 µg/µl trypsin (ca. 1:150 (m/m) trypsin:substrate ratio) at 37 °C. The digestion was quenched by adding 50 µl of 10% v/v formic acid. The resulting tryptic peptides were purified on a 96-well C18-based solid phase extraction (SPE) plate (BioPureSPE Macro 96-well, 100 mg PROTO C18, The Nest Group). The purified samples were resuspended in 120 µl of 0.1% formic acid. 1 µl of OSPP was spiked to 1.5 µg of digested plasma and injected on LC-MS/MS platforms (ZenoTOF 7600, timsTOF, Exploris480) at customized volumes.

**Calibration curves.** We introduce an 8-point calibration curve with BSA as a surrogate matrix. For the seven non-zero calibration samples, 10 µl of the OSPP mixture (the same as what the samples are used for) was mixed with 10 µl of a dilution series of the native peptide standard pool ranging from 1000 to 0.064 pg/µl; 20 µl of BSA tryptic digest was then added as a surrogate matrix. The last sample of the calibration series used 10% (v/v) acetonitrile buffer instead of the light peptide standard (see details in Supplementary Data 3 -Preparation of Calibration curves).

## Liquid chromatography mass spectrometry

**Micro-flow-rate (µflow) LC attached ZenoTOF 7600 (Zeno SWATH DIA, Zeno MRM-HR).** All samples were acquired on an ACQUITY UPLC M-Class system (Waters) coupled to a ZenoTOF 7600 mass spectrometer with an Optiflow source (SCIEX). Prior to MS analysis, 250 ng samples were loaded onto LC and chromatographically separated with a 20 min gradient (time, % of mobile phase B: 0 min, 3%; 0.86 min, 7.1%; 2.42 min, 11.2%; 5.53 min 15.3%; 9.38 min, 19.4%; 13.02 min, 23.6%; 15.48 min, 27.7%;17.27 min, 31.8%; 19 min, 40%; 20 min, 80% followed by re-equilibration for 10 min before the next injection) on a HSS T3 column (300 µm × 150 mm, 1.8 µm, Waters) heated to 35 °C, using a flow rate of 5 µl/min where mobile phases A and B are 0.1% formic acid in water and 0.1% formic acid in acetonitrile, respectively. To avoid introducing technical variance due to differences in injection volumes, we always injected a constant volume of the plasma sample or calibration series samples.

**Zeno SWATH DIA.** A Zeno SWATH DIA acquisition scheme with 85 variable-sized windows and 11 ms MS2 accumulation time was used. Ion source gases 1 and 2 were set to 12 and 60 psi, respectively. Curtain gas was at 25 psi, CAD gas at 7 psi, and source temperature was set to 300 °C; spray voltage was set to 4500 V.

**Multiple reaction monitoring—high resolution (Zeno MRM-HR).** A scheduled Zeno MRM-HR method with identical instrument setting parameters as for Zeno SWATH was developed and used. The choice of precursor and selection of retention time was adopted based on triplicate injections of EDTA plasma sample on the microflow attached Zeno SWATH DIA. The Zeno threshold was set to 20,000 cps and for all peptides, the TOF MS2-scan range was from 200 to 1500 m/z, respectively. MS2 accumulation time was set to 13 ms. Retention time tolerance was set as +/- 20 seconds. Collision energies were defined based on the following formula: CE = slope * m/z + intercept, Supplementary Table 1).

**Analytical flow-rate system LC attached ZenoTOF 7600 (Zeno SWATH DIA MS, ZT Scan DIA).** Samples were acquired on a 1290 Infinity II UHPLC system (Agilent) coupled to a ZenoTOF 7600 mass spectrometer with a DuoSpray TurboV source (SCIEX). Prior to MS analysis, samples were chromatographically separated on an Agilent InfinityLab Poroshell 120 EC-C18 1.9 µm, 2.1 mm × 50 mm column heated to 50 °C. A gradient was applied that ramps from 3 to 36% buffer B in 3 min (buffer A: 1% acetonitrile and 0.1% formic acid; buffer B: acetonitrile and 0.1% formic acid) with a flow rate of 800 µl/min. For washing the column, the flow rate was increased to 1.2 ml/min, and the organic solvent was increased to 80% buffer B in 0.1 min and was maintained for 1.4 min at this composition before reverting to 3% buffer B in 0.1 min. 1.5 µg of the plasma sample or calibration series sample was loaded prior to cohort samples entering MS.

Zeno SWATH DIA acquisition scheme with 60 variable-sized windows and 13 ms MS2 accumulation time was used. Ion source gas 1 (nebulizer gas), ion source gas 2 (heater gas), and curtain gas were set to 60, 65, and 55 psi, respectively; CAD gas was set to 7 psi, source temperature to 600 °C, and spray voltage to 4000 V.

The ZT Scan DIA[72] method used the same instrumental source setup parameters as Zeno SWATHDIA. The method consisted of an MS1 scan from m/z 100 to m/z 1000 and 25 MS2 scans (25 ms accumulation time) with variable precursor isolation width covering the mass range from m/z 400 to m/z 910. Q1 mass width is set as 2.5 Da with a scan speed of 750 Da/s. The applied collision energies were as for Zeno SWATH (derived from a linear equation, see above).

**Analytical flow-rate system LC attached timsTOF HT.** Samples were analyzed on a Bruker timsTOF HT mass spectrometer coupled to a 1290 Infinity II LC system (Agilent). Before MS detection, 5 µg of the

sample were chromatographically separated on a Phenomenex Luna®Omega column (1.6 μm C18 100 A,[73] 30 × 2.1 mm) heated to 50 °C, using a flow rate of 0.5 ml/min where mobile phase A & B were 0.1% formic acid in water and 0.1% formic acid in acetonitrile, respectively. The LC gradient ran as follows: 1% to 36% B in 5 min, increase to 80% B at 0.8 mL over 0.5 min, which was maintained for 0.2 min and followed by equilibration with starting conditions for 2 min.

For diaPASEF MS acquisition, the electrospray source (Bruker VIP-HESI, Bruker Daltonics) was operated at 3000 V of capillary voltage, 10.0 l/min of drying gas, and 240 °C drying temperature. The diaPASEF windows scheme was as follows: we sampled an ion mobility range from 1/K0 = 1.30 to 0.7 Vs/cm2 using ion accumulation times of 100 ms and ramp times of 133 ms in the dual TIMS analyzer, each cycle times of 1.25 s. The collision energy was lowered as a function of increasing ion mobility from 59 eV at 1/K0 = 1.6 Vs/cm2 to 20 eV at 1/K0 = 0.6 Vs/cm2. For all experiments, TIMS elution voltages were calibrated linearly to obtain the reduced ion mobility coefficients (1/K0) using three Agilent ESI-L Tuning Mix ions (m/z, 1/K0: 622.0289, 0.9848 Vs/cm2; 922.0097, 1.1895 Vs/cm2; and 1221.9906, 1.3820 Vs/cm2).

**Nanoflow rate LC attached Exploris 480 (Thermo Scientific).** Samples were analyzed on an Exploris 480 (Thermo Scientific) coupled to a Vanquish Neo UHPLC-System (Thermo Scientific) utilizing a 22 min gradient in nanoflow (0.25 μl/min). For LC separation, the attached column was an in-house packed 20 cm long 1.9 μm column. A shortened gradient time was used with the published acquisition method[74] where mobile phases A & B were 0.1% formic acid plus 3% acetonitrile in water and 0.1% formic acid in 90% acetonitrile, respectively. The LC gradient ran as follows: increased from 2% buffer B to 30% buffer B over the course of the first 14.5 min and increased to 60% buffer B within the next 1.5 min. Finally, buffer B concentration increased to 90% for one min and was held for 5 min to flush the column.

For Orbitrap acquisition, full scans were acquired between 350–1650 m/z with a resolution of 120,000. For MS2 scans, the maximum injection time was set to 54 ms, and scans were made over 40 variable-sized isolation windows.

**Generation of OSPP-specific human spectral library.** A comprehensive spectral library for human stable isotope labeling was constructed through a multistep process using DIA-NN and a custom R script. For all the experiments, we used a project-independent public spectral library, DiOGenes[29] reannotated by Human UniProt[75] (UniProt Consortium, 2019) isoform sequence database (3AUP000005640, [27 March 2023]). The library was first automatically refined based on the dataset at 0.01 global q-value (using the "Generate spectral library" option in DIA-NN). DIA-NN was employed with specific commands to enhance the library's accuracy and utility and label all arginines and lysines in the existing spectral library: --fixed-mod SILAC,0.0,KR, label --lib-fixed-mod SILAC --channels SILAC,L,KR,0:0; SILAC,H,KR,8.014199:10.008269 --peak-translation --original-mods --matrix-ch-qvalue 0.01.

This set of commands facilitated the automatic segregation of the spectral library into multiple channels, particularly for precursors associated with the Lysine and Arginine label group modification. To improve precision and accuracy during quantification, this heavily labeled spectral library was further refined. This refinement involved only keeping the label for peptides from OSPP with only the C-terminal lysine or arginine labeled, and for quantification accuracy, all b-ions were excluded from quantification by labeling b-ions as "T" in the "ExcludeFromAssay" category.

**Raw data processing.** All raw data from the ZenoTOF 7600 system were acquired by SCIEX OS (v. 3.0). All raw data from timsTOF HT were acquired with timsControl (v.5.1.8) and HyStar (v.6.3.1.8). All raw data from Exploris 480 (Thermo) were acquired using Xcalibur.

**Discovery proteomics.** The raw proteomics data from all DIA methods were processed using DIA-NN, 1.8.1, available on GitHub (DIA-NN GitHub repository[76]). The MS2 and MS1 mass accuracies were set to 20 and 12 ppm (ZenoTOF 7600 data) or 15 and 15 ppm (timsTOF and Exploris 480 data), and the scan window to 7. The aforementioned OSPP-specific Human Spectral Library is used for data processing with additional commands: --fixed-mod SILAC,0.0,KR,label --channels SILAC,L,KR,0:0; SILAC,H,KR,8.014199:10.008269 --peak-translation --original-mods --matrix-ch-qvalue 0.01 --restrict-fr --report-lib-info.

Specifically, following a two-step MBR approach[28], an in silico spectral library is first generated by DIA-NN from the FASTA file(s); this library is then refined based on the DIA dataset and subsequently used to reanalyze the dataset to obtain the final results.

The data were filtered in the following way. First, a 1% run-specific q-value filter per isotope channel was automatically applied at the precursor level by DIA-NN (--matrix-ch-qvalue 0.01). We note that in any experiment processed using the MBR mode in DIA-NN, 1% global precursor q-value filtering is also applied automatically[28].

For quantification, we used the "Precursor.Translated" value as quantities for each precursor in MS2 quantification. For Exploris 480 data, since orbitraps are sensitive in MS1, we also used "Ms1.Translated" was used.

**Targeted proteomics.** Zeno MRM-HR data were processed using Skyline (64-bit, v.23.1.0.268). No blinding was performed during peak integration. The relative quantity of each peptide is calculated by the summation of peak areas of each selected fragments of a peptide (fragments used in DIA calculation extracted from DIA-NN output; the list of fragments used for quantification in Supplement Data 5).

**Calibration curve, LOD, LOQs.** The calibration curve (fixed amount of OSPP mixed with serial dilutions of native peptide standards in BSA (4 ng/μl), covering a concentration range of ~ $2 \times 10^5$ (details on the preparation of the calibration curve is in Supplementary Data 3—Preparation of Calibration curves) was measured in technical triplicates on each of the LC-MS/MS systems for each of the 211 peptides.

For limit of detection (LOD) calculation, blanks showed negligible signal; therefore, the standard deviation of the y-intercepts from replicate calibration curves was used to estimate LOD. The LOD was calculated as: LOD = 3.3×σ/S and the calculated LOQ as LOQ = 10 × σ/S, where σ is the standard deviation of the y-intercepts across replicates, and S is the mean slope of the corresponding calibration curves. This approach ensures that the LOD reflects assay sensitivity even in the absence of measurable blank noise. Calculated LODs for peptides, including CRP-derived peptides, were below the reported reference range for healthy adults, indicating that the assay is sensitive enough to detect physiologically relevant concentrations. for the detectability reported in the manuscript, "detectable" refers to a detected peak with S/N > 3 by manual inspection.

The upper and lower limit of quantification (-LLOQ and −ULOQ) was determined based on the accuracy of replicated injections (n = 3) on the same LC-MS platform (CV < = 20% or expanded to CV < = 40% if less than 4 calibration point are presented in previous criteria). Peptide concentration (expressed in pg/μl) was determined from calibration curves constructed with native and isotopically labeled peptide standards in the surrogate matrix (4 ng/μl BSA) and manually inspected and validated. Peptide values below the lowest or above the highest detected calibrant concentration across all samples were removed from the analysis. Linear regression analysis of each calibration curve in each acquisition platform was performed using custom R code (with 1/x weighting).

**Data analysis.** All quantitative data presented and used for statistical analysis are in linear scale and were not log-transformed at any stage of the analysis.

**Data completeness**. The completeness of data for each peptide was evaluated based on its frequency of detection across all biological samples. Peptides were considered if they were detected in more than 66.7% (⅔) of the samples. We calculated the percentage of each peptide measured on each LC-MS platform/method and used only the peptides with a completeness value exceeding 66.7% for subsequent analysis.

**Data normalization**. Two normalizing strategies to evaluate the quantification consistency were applied. The first approach was a normalization by median division of all endogenous peptide quantities in the study pools (except in the Thermo instrument, replicate 04 is excluded due to acquisition failure) measured on each platform. All peptides in each platform were applied with this factor, referred to as "norm_light".

median(MS) = median(light(MS), na.rm = T)
norm_light = light / median(MS)

In addition, with the spiked OSPP mixture, we use the heavy isotope-labeled spiked peptide standard in each sample to normalize the corresponding endogenous peptide levels in the sample (endogenous peptide quantity (light) / quantity of correspondent heavy labeled peptide (SIS)), termed as "ratio".

ratio = light / SIS

**Precursor selection**. As several precursors (charge state of +1 to +4) from the same peptide are quantified on different platforms, several criteria should be fulfilled to choose the best precursor used for follow-up quantification and cross-method/cross-platform comparison: a) Due to the difference in analyte ionization ability on various MS platforms, different precursors from the same peptide will show various abundances; the most abundant one shall be the charge state with the best ionization efficiency. b) Moreover, the abundance will also affect the reproducibility of the performance of isotopically labeled peptide standards. To replicate injections of study pool samples on each platform, we filtered for precursors with CV less than 40% to guarantee reproducibility. c) Additionally, for precursors of different peptides from the same protein, we checked the behavior of isotopically labeled peptide standards throughout all study samples and only chose the precursor that showed the same trend. The precursors used for quantification on different MS platforms are listed in Supplementary Data 7−Precurs used for quantification.

## Statistical analysis

**Wilcoxon signed-rank test.** To assess whether the relative abundance rankings of peptides differ across blood matrices, we performed pairwise Wilcoxon signed-rank tests for each peptide. For each comparison, peptide intensity values from matched samples (same donor measured across matrices) were tested using the wilcox.test() function in R (v4.2.2) with paired = TRUE. All six pairwise matrix combinations were tested independently. To account for multiple hypothesis testing, p-values were adjusted using the Benjamini-Hochberg (BH) procedure with the p.adjust(method = "BH") function. The resulting adjusted p-values were compiled into a long-format data table with peptide identifiers and matrix comparison labels. Peptides showing significant rank shifts (adjusted $p < 0.05$) were flagged for further interpretation.

**Coefficient of variation (CV).** To assess intra-group variability of peptide precursors, we calculated the coefficient of variation (CV) using two methods. The primary method employed a robust measure, defined as the median absolute deviation (MAD) divided by the median of intensity values, expressed as a percentage (CV = MAD/Median × 100). This approach was implemented using the mad() function from the R stats package (v4.2.2), offering robustness to outliers and suitability for skewed or heteroscedastic data commonly observed in

proteomics. For comparison, we also computed CV using the conventional formula based on standard deviation divided by the mean (CV = SD/Mean × 100). Both measures are reported where relevant to provide complementary insights into variability.

**Kendall's tau (KT) trend test.** Significance testing of the trend between peptide quantities (normalized endogenous quantity or ratio) and the ordinal classification as provided by the WHO disease severity (levels as indicated) was performed using Kendall's tau (KT) statistics as implemented in the "EnvStats v2.8.1" R package "kendallTrendTest" function. For the clinical cohort, the KT statistics were calculated as the trend of absolute peptide concentrations against the following WHO groups: 0, 3, 4, 5, 6, and 7:

EnvStats::kendallTrendTest(value ~ as.numeric(I.WHO), data = data)

or calculated as trend of absolute peptide concentrations with respect to disease severity (treated as an ordinal factor with the levels healthy <mild <severe <critical, which was converted to numeric ranks prior to analysis):

Severity_num = as.numeric(factor(
F.Severity,
levels = c("healthy", "mild", "severe", "critical"),
ordered = TRUE))
EnvStats::kendallTrendTest(value ~ Severity_num, data = data)

Selected peptides in each comparison were used for data analysis, without imputation. Kendall's rank correlation (Kendall's τ) was used to assess monotonic trends in peptide intensities across disease-severity groups. False discovery rates (FDR) were calculated using the Benjamini−Hochberg procedure. Because each mass-spectrometry (MS) platform represented an independent experiment, FDR correction was performed within each MS dataset (and type) separately rather than across all datasets combined. Multiple testing correction was performed by controlling for false discovery rate using the Benjamini-Hochberg procedure 1 as provided by the R package "stats v4.2.2"−"p.adjust" function. A full summary of these statistical test results is provided in the respective supplementary Data. (Adjusted) p-values were considered significant when $p < 0.05$.

**Intraclass correlation coefficient (ICC) analysis.** Inter-platform reproducibility was assessed using intraclass correlation coefficients (ICC). For each precursor−fragment−label group, ICC(3,k) was computed with the psych package (version 2.5.6) using a two-way mixed-effects model to evaluate the reliability of the average measurement across MS platforms while allowing missing values. Pairwise platform agreement was additionally quantified using ICC(2,1), calculated for every platform pair with the irr package (version 0.84.1), and the resulting ICC values were averaged to summarize pairwise reproducibility.

**Variance partitioning analysis.** To assess the contribution of technical and biological factors to peptide-level variability, we used the variancePartition R package. Variance partitioning was applied at the peptide level, which estimates the proportion of variance in peptide intensity explained by each variable (e.g., disease severity, MS batch). The resulting variance fractions were summarized across all peptides to assess the overall contribution of biological versus technical sources of variation. As we only focus on the variance brought by different LC-MS platforms, a linear mixed-effects model was only applied to both MS platform (MS) and disease severity (Severity) as random effects, no other biological variance (e.g. age, sex, BMI) was included in the analysis, the formula used is shown as:

$$expression \sim (1|MS) + (1| Severity)$$

Missing values in the peptide intensity matrix were imputed using k-nearest neighbor (KNN) imputation ($k = 10$) with the impute (v 1.80.0) R package. Prior to imputation, the expression matrix was transposed so that samples were treated as observations and peptides as variables, and subsequently transposed back to the original orientation after imputation.

The model was fitted using fitExtractVarPartModel() from the variancePartition v1.36.2 R package, and the proportion of variance explained by each factor was computed for each peptide. The results were summarized to quantify the overall impact of technical and biological variance in the dataset.

**Bland–Altman analysis.** To assess the agreement between different mass spectrometry acquisition methods, we performed Bland–Altman analyses at the peptide level (using ratios in two datasets). For each experimental condition, peptides were filtered to exclude infinite values and missing data. For every peptide detected under each condition, the mean and difference of the ratios from two datasets (e.g., hZsSWATH dataset and the composite dataset (not including any data from hZsSWATH dataset to avoid data leakage)) were calculated. The bias (mean difference), standard deviation (SD), and limits of agreement (LoA, calculated as bias $\pm 1.96 \times$ SD) were determined. Confidence intervals (95%) for the bias and LoA were estimated using Student's t-distribution and the standard errors derived from the sample variance and size. Bland–Altman plots were generated for each peptide, visualizing the agreement between methods along with the calculated LoA and confidence intervals. The corresponding summary statistics were compiled into Supplementary Data 7—Bland-Altman output. All calculations and visualizations were performed in R using custom scripts based on the tidyverse (v 2.0.0) package for data handling and base R graphics functions for plotting.

**Data visualization.** To visualize the overlap of peptides (or peptide groups) across different experimental conditions or statistical criteria, set visualizations were generated using both UpSet and Venn diagram approaches. UpSet plots were created using the upset() function from the UpSetR v1.4.0 R package. For comparisons involving two to four sets, Venn diagrams were generated using the ggvenn() function from the ggvenn v0.1.10 R package. All other visualization is performed using "ggplot2 v.3.5.1".

Principal component analysis was performed and visualized using the fviz_pca_biplot function from the factoextra v1.0.7 R package. Missing values in the peptide intensity matrix were imputed using k-nearest neighbor (KNN) imputation ($k = 10$) with the impute (v 1.80.0) R package. Prior to imputation, the expression matrix was transposed so that samples were treated as observations and peptides as variables, and subsequently transposed back to the original orientation after imputation. Biplots display both individual samples (observations) and peptides (variables) in the principal component space, allowing assessment of sample separation according to severity group and identification of key peptides contributing to principal component loadings. Points were colored by clinical severity category (e.g., healthy, mild, moderate, critical) or acquisition platforms, and variable arrows represent the contribution and direction of individual peptides in the PCA space. Ellipses representing the 95% confidence interval (ellipse.level = 0.95) around each acquisition platform.

**Reporting summary**
Further information on research design is available in the Nature Portfolio Reporting Summary linked to this article.

## Data availability
All metadata regarding the clinical studies used in this manuscript were provided, individual de-identified participant data (including data dictionaries) are included, for COVID-19 study cohort, only the auxiliary sample name and respective WHO grade and severity group information are shared in Supplementary Data 4—meta data for study cohort; for matrix test, age range and sex of individuals are provided in Supplementary Data 4—metadata for 4 matrices. The raw proteomics data generated in this study have been deposited in ProteomeXchange with identifier PXD070765. The processed proteomics data are available at Mendeley[77]—Proteomics Data (DIA-NN & skyline). All data generated in this study and used for manuscript and plotting are provided in the Supplementary Data. LC-MS acquisition schemes, data analysis pipeline, spectral library are available online in Mendeley[77]—Data Processing pipeline & spectral library; Source data (data processed by DIA-NN/ skyline) used for data analysis and visualization are provided in Mendeley[77]—Proteomics Data (DIA-NN & skyline). Source data are provided with this paper.

## Code availability
All codes used in the manuscript are available in Mendeley[77]—Code Available.

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

## Acknowledgements

We thank all members of Charité Core Facility High Throughput Mass Spectrometry and Lei Feng (Institute of Forensic Science, Ministry of Public Security, China) for technical support in the preparation of peptides and also Arturas Grauslys (Eliptica) for helping with data analysis. We thank Prof. Dr. Daniel Teupser from the Institute of Laboratory Medicine, Munich, Germany for providing us a list of proteins with clinical chemistry assay. We further thank the Pa-COVID-19 study group[37,39] for study logistics and collection of biosamples and clinical data. This work was supported by the Ministry of Education and Research (BMBF), as part of the National Research Node 'Mass Spectrometry in Systems Medicine' (MSCoreSys), under grant agreements 16LW0239K (to M.M.), 01EP2201 (to M.R.); and the Deutsche Forschungsgemeinschaft (DFG, German Research Foundation) for grant no. 492697668 (to M.M.); and under Germany's Excellence Strategy – EXC 3118/1 – project no. 533770413 (to M.R); and the European Research Council (ERC) under grant agreement ERC-SyG-2020 951475 (to M.R.). This work was further supported by a BIH Booster Grant (2022-B3020086-11/12 to Z.W., P.T-L., F.K., J.H., M.M., M.R.) and DKTK under grant agreement BE01 1020000483. Z.W. was a member of the International Max Planck Research School (IMPRS) for Infectious Diseases and Immunology and the DFG-funded Sonderforschungsbereich (SFB) TRR 186 and SFB1588. Figure 1 was created in BioRender (WANG, Z. (2025) https://BioRender.com/8az8cdz).

## Author contributions

Z.W.: Experimental Design, Data Curation, Sample Preparation, Data Collection, Methodology, Formal Analysis, Visualization, Writing—Original Draft Preparation, Writing—Revision. V.F.: Formal Analysis, Peptide Selection, Writing—Original Draft Preparation, Writing—Revision. LR.S: Data Collection, Visualization, Writing—Original Draft Preparation, Writing—Revision. P.T-L.: Sample Collection, Writing—Original Draft Preparation, Writing—Revision. D.L.: Sample Preparation. F.A.: Data Collection. K.T-T.: Data Collection. A.F.: Data Collection. A.N.: Sample Preparation. AS.W.: Data Collection. AAJ.W.: Data Collection. L.L.: Data collection. F.K.: Consultation, Funding Acquisition. M.S.: Consultation, Supervision. J.H.: Conceptualization, Funding Acquisition, Supervision, Consultation, Writing—Original Draft Preparation. M.M.: Conceptualization, Funding Acquisition, Project Administration, Resources, Supervision, Consultation, Writing—Original Draft Preparation, Writing—Revision. M.R.: Conceptualization, Funding Acquisition, Project Administration, Resources, Supervision, Consultation, Writing—Original Draft Preparation, Writing—Revision.

## Funding

## Competing interests

Markus Ralser is founder and shareholder, Luise Luckau was an employee, Ziyue Wang, Michael Müellder were/are advisors of Eliptica Ltd. The remaining authors declare no competing interests.

## Additional information

[1]Department of Biochemistry, Charité–Universitätsmedizin Berlin, Corporate Member of Freie Universität Berlin and Humboldt-Universität zu Berlin, Berlin, Germany. [2]Core Facility—High-Throughput Mass Spectrometry, Charité—Universitätsmedizin Berlin, Corporate Member of Freie Universität Berlin and Humboldt-Universität zu Berlin, Berlin, Germany. [3]Department of Infectious Diseases and Critical Care Medicine, Charité—Universitätsmedizin Berlin, Berlin, Germany. [4]Proteome Dynamics, Max Delbrück Center for Molecular Medicine, Berlin, Germany. [5]Faculty of Life Sciences, Humboldt-Universität zu Berlin, Berlin, Germany. [6]Eliptica Limited, The London Cancer Hub, London, Sutton, UK. [7]Berlin Institute of Health (BIH) at Charité—Universitätsmedizin Berlin, Berlin, Germany. [8]The Wellcome Centre for Human Genetics, Nuffield Department of Medicine, University of Oxford, Oxford, UK. [9]Max Planck Institute for Molecular Genetics, Berlin, Germany. [10]These authors jointly supervised this work: Michael Mülleder, Markus Ralser.
✉e-mail: michael.muelleder@charite.de; markus.ralser@charite.de

