## [Transparent Peer Review file · Nature Communications]

Cross-platform Clinical Proteomics using the Charité Open Standard for Plasma Proteomics (OSPP)

Corresponding Author: Professor Markus Ralser

Version 0:

Reviewer comments:

Reviewer #1

(Remarks to the Author)

Summary:

Wang et al. present an interesting work, where they developed the Charite Open Peptide Standard for Plasma Proteomics (OSPP), a panel of 211 peptides that can be used for both targeted and untargeted proteomics. This is indeed a quite laborious work of plasma proteomics method development. The authors further demonstrate to some degree a potential for clinical utility in a small cohort of COVID-19 patients, where they demonstrate a continuous trend in differential levels of plasma proteins associated with COVID-19 severity.

The authors' efforts in providing an open resource that includes good quality peptides are much appreciated. However, there might be some biases related to the peptide selection that the authors should clarify. From a proteomic perspective, the manuscript misses to explain what the OSPP adds to the field of targeted proteomics that previous efforts have not.

The claims regarding the OSPP's demonstrated clinical utility in patient classification and biomarker identification are too strong and not supported by the evidence presented in the manuscript. The authors should be more precise in their language regarding the clinical potential of the panel. Considering the proteins included in the panel, it is not quite clear if the panel would have a broader clinical potential. The panel might have applications in acute, systemic inflammatory conditions but even for this case the authors need to argue further what would be the added value in comparison to other panels and assays, some of which are widely used at the clinic.

There are also some concerns in data analysis and the presentation of methodology and findings, which the authors must address. There were many typos and lack of clarity in some parts of the manuscript, with the latter limiting the understanding of some parts in the results and methods section. Under specific remarks, I point out some examples. The authors should carefully correct the language, clarify their statements, and mind the tense when reporting results.

Specific comments:

1. Page 4, Lines 14-28 and Page 22, Lines 5-15: Please recheck the mathematical annotations of the selection algorithm. Some elements are not well defined, and some expressions appear to be written in programmatic language (e.g. $==$). It is not clear in which units were the ranks expressed and how were the relative ranks calculated. Was the $CV(p,n)$ calculated on log-transformed or raw values?
2. It would be useful to the reader to specify whether during selection there was a condition that some project characteristics had to be considered or $\text{PresInNProjects}(p) \geq 4$ could mean that all four projects are EDTA-plasma. Would be useful to the reader to report which peptides were present in which project, to infer whether some are specific to a sample type. Apologies if I missed that information elsewhere, but I couldn't find it in Supplementary Table 1.
3. How did the sample type bias the ranks and relative ranks? Can this be estimated? This could be relevant for the coagulation factors or other proteins that are being captured in the coagulum. It would be relevant to explain this in the manuscript. It would be relevant to further explain the setting of blood sample collection and the processing that led to obtaining the plasma/serum samples.
4. How did the weighing approach bias the selection? E.g. if the maximum presence of a peptide was let's say 50% in a study and the same peptide had very low CV of 1%, the weight would have been the same as for a peptide that had 100% presence and 2% CV. Both CVs would have been low, but the latter peptide would have been detected in twice more samples. Would these two peptides be equally relevant for downstream selection, i.e. what is the rationale that the presence and CV should have equal importance in the weighing?
5. Why was the lower cut-off of the relative rank average set to be 0.6?
6. It would be important to specify in Figure 1 the stepwise exclusion of peptides at each step. The authors should specify

the total number of peptides per project, followed by the number of peptides that are excluded at each step when a criterion is not met (e.g. how many peptides were excluded because they were not proteotypic or did not meet the chemical properties criteria, etc.)

7. Are some proteins represented by a single peptide? How has this affected the accuracy and precision in quantifying the different proteins? It would be useful to present the effect of the selection-related factors on precision in supplementary.

8. Figure 2b – from where were the annotations obtained in the second, third and fourth panel from above?

9. How was the limit of detection (LOD) determined? For proteins that have low abundance in non-diseased blood samples (such as CRP) – did the assay measure concentrations with the LOD below the reference values cut-off for that protein in a healthy population (where those are known) or was the LOD above the reference cut-off? This is important, since if the assay is to be clinically relevant, it should be able to measure concentrations that would include reference values as well.

10. Page 8, Lines 35-38: the COVID-19 cohort is not well balanced. It has a different number of patients with different levels of severity. I assume effective size is a typo referring to effect size. What does effect (or effective, if not a typo) size refer to in this sentence?

11. It is unclear how the design allows for “identification of peptides that are only abundant in either healthy or diseased samples, avoiding missing values due to low abundance”. Was there a pattern of missing values for some proteins that was related to the COVID-19 status? E.g. CRP levels would be elevated in a good portion of COVID-19 patients but not in some very mild cases or in healthy individuals. Shouldn't the latter two groups have very low-abundant CRP levels that might be in some cases undetectable? Or was it undetectable regardless of the COVID-19 status? Same can be said for many of the other proteins for which there is strong evidence that they are elevated in COVID-19, like ORM1 and ORM2. The authors should explore if the missingness in the data is related to COVID-19 status.

12. There is a lack of clarity/consistency in the terminologies regarding the WHO severity scales for COVID-19. Seems that the WHO ordinal scale used in the study refers to the scale from reference 41 and not reference 42. The WHO ordinal scale has 8 orders (8 = death) – should be corrected in the manuscript. The reference 42 refers to the WHO clinical progression scale. In Figure 2, the authors write “WHO treatment escalation score”. It is important to clarify which scale exactly was used, to use the right terminology consistently, and to cite it properly.

13. Although the focus of the study is on plasma proteomics method development, the cohort of COVID-19 patients needs further description – at least what is the distribution of age and sex because these factors can affect the plasma protein levels. How have age or sex biased the analysis of the association between protein levels and COVID-19 severity, since it is well known that male sex and older age are associated with severity and worse outcomes in COVID-19 patients?

14. Figure 3b: PCA is not clustering. What do the ellipses represent? Information on these analyses are missing in the methods section too.

15. Page 11, Line 34: I assume the authors refer to Figure 3c. Why were the changes in protein levels normalised to WHO grade 3 patients? Wouldn't it be a more intuitive comparison to normalise the fold changes to healthy controls?

16. Figures 3c and 3d nicely present a gradual change related to COVID-19 severity. However, the authors should further explain why C3 and CST3 are presented in Figure 3d, since the published evidence for their elevated levels in COVID-19 patients is variable. Wouldn't it be more relevant to prioritise other proteins for presentation? The authors should include more references related to the claims on Page 11, Lines 34-45.

17. Supplementary Table 5. Would it be valid to use the Kendall's Tau trend test for an analysis of trend where the orders are not proportional? In the analyses, how were the patient groups treated numerically – as orders of 0, 3, 4, 5, 6, and 7, or from 0 to 5, or as grouped based on the severity group as 0 (healthy), 1 (mild), 2 (moderate), and 3 (critical)? The authors should explain the rationale for their analytical decision and how a different categorisation of the order could have affected the analysis.

18. The boxplots show data related to the groups described in Figure 3a. They should be presented in the same manner the orders are defined in the trend analysis if the authors justify the validity of the method. It can be helpful to the reader to add p values for the comparison in the boxplots. How have age and sex affected the association between the plasma proteins and COVID-19? The protein and gene names that correspond to the peptides should be included in the supplementary tables in all sheets that show peptide information.

19. How consistent is the association between A2M and COVID-19 based on published evidence?

20. Page 16, Lines 35-40: The % related to the variance explained differs from the one shown in Figure 5c. Please recheck. In the format it is presented, it is not clear from the PCA that the primary source of variance is related to biology.

21. Page 17: the use of the term “composite cohort” is misleading. Based on the information presented in the manuscript, the cohort is not a separate cohort nor a separate MS run but a computationally generated dataset for the purpose of comparing the agreement in quantifications of the OSPP between a single platform and a hypothetical scenario where the sample could have been analysed on any of the platforms. If that is correct, the authors should explain this clearer and avoid expressions like “two cohorts”. The authors should also compare the agreement between each of the single platforms and the agreement between the randomly sampled dataset and the remaining platforms, to demonstrate that the high agreement is reproducible when choosing any of the other single platform runs. When generating the randomly sampled dataset, the authors should consider excluding the single platform they intend to compare the randomly sampled dataset to, since sampling from the dataset of the comparing platform would imply a leakage of information, overestimating the agreement.

22. Page 17, Lines 19-25: The Bland-Altman plot analysis is unclear. Is the quantification consistency measured in two cohorts or in two different runs on the same samples from the same cohort on another platform (see comment above)? Figure 5f is presented in relation to the claim that the quantification between the two measurements agree nicely but it represents a single peptide (from A2M) and not the agreement in all peptides. Lines 23-25: This segment refers to the agreement in quantifications of all the peptides and should be presented as a main figure instead. The bias close to absolute values of a difference of 1 in some proteins is not negligible.

23. The claim on page 19, Lines 29-31 is not supported by the presented evidence. The OSPP is not validated for patient stratification and outcome prediction. The authors have demonstrated that some proteins have a gradual increase or decrease in levels at the time of a given COVID-19 severity. OSPP is not built into a model that provides patient stratification and outcome prediction.

24. Were the DIA data log-transformed prior to normalisation?
25. Why was the CV calculated based on median and median absolute deviation and not mean and standard deviation? Did this approach underestimate the CVs?
26. Some analyses were performed but were not described under statistical analysis and visualisation.
27. Some examples of typos that the authors should correct are provided below. There were more – the authors should carefully recheck the manuscript.
- Page 3, Lines 31-33: maybe these two sentences were supposed to be connected by a comma?
 - Page 4, Line 7: there is an additional “from” in front of prepared.
 - Page 8, Line 1: “in combination” with is missing?
 - Page 8, Line 20: sentence unclear.
 - Page 11, Line 30: a closing bracket is missing. And the subsequent sentence is unclear and long.
 - Discussion: the second sentence is long and unclear, with some typos.

Reviewer #2

(Remarks to the Author)

The manuscript by Ralser and coworkers presents the development, validation, and application of the Charité Open Peptide Standard for Plasma Proteomics (OSPP). This is an open-source and cost-effective stable isotope-labeled internal standard designed to improve precision, reproducibility, and cross-platform comparability in plasma proteomics workflows. The peptide selection process is methodical, combining empirical data from thousands of plasma proteomes and control injections. The use of a relative rank metric ensures the selection of analytically robust peptides with low technical variability. The validation of the OSPP across different platforms (e.g., ZenoTOF 7600, timsTOF, and Orbitrap Exploris 480) demonstrates its versatility and robustness. The authors demonstrate the utility of OSPP for both targeted and untargeted proteomics across multiple LC-MS platforms in a case study involving a COVID-19 patient cohort. The manuscript is easily readable and highlights an affordable technique that could help standardize plasma proteomics. However, I have a number of comments and concerns that need to be addressed before I can recommend publication.

Specific comments to address:

I assume that all the proteotypic peptides selected for the OSPP standard are tryptic peptides but this is not specified in the manuscript text. Were the selected peptides derived from other proteases than trypsin or are they all fully tryptic peptide sequences? This should be clearly explained in the manuscript text.

The peptide/protein relationship is an issue that should be more directly addressed. The authors are very open that they are only evaluating peptide quantification but the abstract and the title reflects that they are also looking at protein quantification even though this is not assessed in the manuscript. This should be addressed. Preferably the potential on protein level quantification should be evaluated and if not the abstract should reflect this and the text should discuss the connection. Orthogonal measurements (eg. ELISA or RIA) of at least some proteins would also be beneficial.

While the OSPP includes 211 peptides from 131 plasma proteins, this represents only a small subset of the plasma proteome. Key biomarkers or protein isoforms relevant to specific diseases may be missing, limiting its applicability in certain contexts. To demonstrate the advantages and also the limitations of the OSPP standard, the authors should benchmark the performance of the OSPP standard against the state-of-the-art peptide standards for plasma proteomics, i.e. the PQ500 standard from Biognosys. This benchmark should be performed evaluating plasma proteomics from a clinical cohort in terms of biomarker coverage and peptide and protein quantitation.

It is great that the authors evaluated different plasma/serum material and different MS setups. Additionally different protein digestion protocols or different centers would also be very relevant to test the true potential for standardizing. It should at least be discussed as a limitation.

The OSPP was primarily validated in the context of a COVID-19 cohort. While this is a relevant use case, additional studies in other diseases (e.g., cancer, cardiovascular diseases) would strengthen the generalizability of the standard for broader clinical applications.

Page 4, line 17: It is a good strategy to select peptides based on previous knowledge of detection and reliable quantification with LC-MS. The strategy will be biased by the samples included in the datasets. And if the criteria deselect peptides with low CVs in a database using non-healthy and healthy samples, then peptides with biological dynamics in disease states could be filtered out even though they might be well quantified. Similarly, the resulting list includes several IgG-related genes that might be of questionable value from a biomarker perspective. Is there a rationale for including so many IgG peptides?

Page 8, line 1-14: The robustness test across different plasma/serum material is very relevant. There is no figure to illustrate these data. Should be included – at least supplementary. Information regarding the material handling (centrifugation steps) is lacking.

Page 14, line 12: Why is A2M chosen as example? It would be nice to see examples of differential abundance. E.g. a peptides from each “concentration bin”.

Page 15, figure 5c: The PCA plot is central to the result section, and I do not think the technical vs. biological accuracy is very clearly shown.

Page 19, line 4: "Indeed, the use of an internal standard can also be helpful to improve the 5 quantification of peptides that are not covered by the standard". This is potentially a strength of the standard but is not evaluated in the text nor is there any references to back up the claim. The text should reflect this or find references.

Page 19, line 8: "simplifying data analysis is an underestimated benefit of SIS panels". It does not appear that the overall data analysis will be more simplified with SIS. It requires specific software and there are problems regarding peptide detection etc., so it could in some cases also make it more complex.

Reviewer #3

(Remarks to the Author)

This paper logically lays out the selection of 211 tryptic peptides that arise from plasma proteins and the development of these peptides into a standard "kit" that can be used across both targeted and untargeted (DIA) mass spectrometry-based plasma studies. The authors provide an example of its application in a small COVID-19 patient study. Finally, they provide the information needed for the inclusion of all or some of these peptides in the spectral libraries that would be needed from transferability.

This is a very well-designed approach, and it has been exceptionally well-executed. The ability to have a set of SIL peptides in which the performance has been well-characterized and that reach the preclinical requirements will allow, if adopted, cross-comparison between mass spectrometry platforms and workflows, different laboratories and, importantly, act to allow normalization between cohorts.

Important considerations.

1. It would be very helpful if the authors included a dynamic range figure that shows the concentration range of each endogenous peptide and the concentration of the corresponding SIL peptide.
2. The authors need to include the retention times and the retention time variability for each peptide, at least across the Covid study. If possible, also in the study using the Exploris 480 instead of the ZenoTOF 7600. This is important as this will impact whether these peptides can be used as retention time standards (replace IRT in DIA studies) as well as quantitative internal standards.
3. The authors discuss potential expansion of the multiplex but also need to state (speculate) the minimum number of peptides needed for the SIL reproducible/cross valuation kit. This can be part of a "next stage/limitation section," which could include deployment across more instruments and laboratories. The authors may also want to address how this could be used for comparison to other proteomics studies that measure protein/proteofom using capture (or other) reagents.

Minor but important.

1. The paper should not include cost-effectiveness as there is no cost associated with the kit yet, and this may not be the case. Please remove this from the manuscript.
2. The protein full name is cardiac troponin T (not just troponin T, as there are several isoforms) and should also include cardiac troponin I, which is equally used clinically.
3. Figure 5C is hard to read and should go in the online supplement, where it can be enlarged.
4. There are a few typos that need to be fixed.

Version 1:

Reviewer comments:

Reviewer #1

(Remarks to the Author)

Summary:

The authors have addressed most of the raised concerns and improved the text. I only have two important concerns remaining: a) the choice of some exemplary proteins to demonstrate the utility of the panels, particularly A2M; and b) the choice and design of statistical models. I believe the authors are well equipped to address these. Another important suggestion, though not one that should determine acceptance, concerns the supplementary tables – better descriptions and/or text references would benefit readers.

Specific comments :

1. Figure 3b: Please note that PCA is not a method for clustering, but a method for dimensionality reduction. It should not be referred to as unsupervised clustering, as done so in figure legends. Ellipses are not confusing. Using ellipses with confidence intervals (CI) to show the spread and covariance structure of each group in a geometric space is excellent (95% CI preferred than 80%), but it's important to explain which method was used to calculate those areas. This information should be specified in the methods section.

2. Figures 3c and 3d: The issue with C3 and CST3 is not because they are understudied. On the contrary, if the reviewers screen the literature, they will find many studies that show inconsistent association with COVID-19 status. Shouldn't in the context of this study, the utility of the panel be demonstrated with regards to well-known changes in plasma protein levels in COVID-19 patients.

3. Figure 3 and text. On several occasions the authors write that they have performed ordinal linear regression, whereas under methods they describe ordinal logistic regression. These two methods are different and imply different assumptions. Please recheck your code to ensure that you have used ordinal logistic regression and correct the text. Related to this – does ordinal logistic regression require proportionality in the plasma protein change linked to COVID-19 severity?

4. To my previous question about how consistent the association between A2M and COVID-19 is based on published evidence, the authors respond that there was a decline in severe COVID-19 in previous works and refer to references 49 and 50. Based on the text under results, I assume that the authors refer to references 48 and 49, since reference 50 is not related to the question. However, reference 48 is an old review from 2021 that concludes that “analysis of A2M protein levels in plasma samples from patients with COVID-19 revealed no significant differences or correlations with other disease parameters”. Reference 49 is a transcriptomics study that finds decrease in A2M transcript levels in peripheral blood mononuclear cells. Perhaps the question was not specific enough to imply that the evidence in question is related to plasma proteomics data. The authors should still explain why A2M is highlighted as an exemplary protein to demonstrate the panel's utility for COVID-19 severity. I would further recommend evaluating the published literature on the relevance of this protein and reconsider highlighting more consistent proteins. In addition to the inconsistent published evidence, the authors highlight that A2M, an (moderate) acute phase protein decreases in severe COVID-19 patients. It is counterintuitive in a disease that has high systemic inflammation.

5. Adding a variance partitioning analysis is very useful. However, the supplementary table was poorly described. The authors need to clarify why the variance partitioning was performed per peptide and not on the entire dataset if they wanted to summarise the effect of biological signal vs. technical bias. Furthermore, the model is too simplistic to include only severity and MS as variables. The authors should consider including age, sex, site, sample type, and other potential confounders, and/or explain why other factors have not been included in the model. The authors then write under methods that a linear mixed-effects model was applied, but this makes the results unreliable since severity is on an ordinal scale, and not a proportional one (see question above). The authors should reconsider their model choice and use a model that is fitting for the used variables or explain how this model is applicable to the dataset. Last, the authors write that “missing values are imputed with zero” – please specify whether this refers to proteomics data or clinical data. In this setting the values should not be imputed for clinical data. For proteomic data, imputing with zero will severely reduce the variance in the data and a multiple imputation model should be considered.

6. This also leads to me an additional point – the supplementary tables are not well organised and require improvement in structure. There are many sheets under a single Supplementary Table, and these should be referenced separately. E.g. the variance partitioning table was referenced as Supplementary Table 7, but it was on sheet 15 and the structure of the spreadsheets was not helpful in finding the right information.

Reviewer #2

(Remarks to the Author)

The authors have satisfactorily addressed by comments and concerns in the revised version of their manuscript. I have no more comments.

Reviewer #3

(Remarks to the Author)

The authors have done a great job in addressing my concern. I particularly like the inclusion of the dynamic range figure. My only extremely minor but extremely important point is that the authors need to change gender to sex. Sex is the correct term as gender includes other factors than biological sex, such as social status, which is not what is captured in this data. Please make this change in Supplemental Figure 2 and throughout the manuscript. I missed this in my first review.

Version 2:

Reviewer comments:

Reviewer #1

(Remarks to the Author)

The authors have thoroughly addressed all the suggestions and concerns. I thank them for their patience and look forward to sharing this interesting work.

Point by point reply_ Wang et.al. -OSPP

Reviewer #1 (Remarks to the Author)

Summary:

Wang et al. present an interesting work, where they developed the Charite Open Peptide Standard for Plasma Proteomics (OSPP), a panel of 211 peptides that can be used for both targeted and untargeted proteomics. This is indeed a quite laborious work of plasma proteomics method development. The authors further demonstrate to some degree a potential for clinical utility in a small cohort of COVID-19 patients, where they demonstrate a continuous trend in differential levels of plasma proteins associated with COVID-19 severity.

The authors' efforts in providing an open resource that includes good quality peptides are much appreciated. However, there might be some biases related to the peptide selection that the authors should clarify. From a proteomic perspective, the manuscript misses to explain what the OSPP adds to the field of targeted proteomics that previous efforts have not.

The claims regarding the OSPP's demonstrated clinical utility in patient classification and biomarker identification are too strong and not supported by the evidence presented in the manuscript. The authors should be more precise in their language regarding the clinical potential of the panel. Considering the proteins included in the panel, it is not quite clear if the panel would have a broader clinical potential. The panel might have applications in acute, systemic inflammatory conditions but even for this case the authors need to argue further what would be the added value in comparison to other panels and assays, some of which are widely used at the clinic.

There are also some concerns in data analysis and the presentation of methodology and findings, which the authors must address. There were many typos and lack of clarity in some parts of the manuscript, with the latter limiting the understanding of some parts in the results and methods section. Under specific remarks, I point out some examples. The authors should carefully correct the language, clarify their statements, and mind the tense when reporting results.

We thank the Reviewer for their detailed report and the constructive comments. We have addressed all specific comments below. To address the point about clinical utility, we apologise that our attempts to demonstrate different levels of application of a broad internal standard in plasma proteomics created confusion. The OSPP is designed to be an internal standard, and the result of an untargeted, disease-agnostic selection of peptides with ideal analytical properties. The principal application of such a panel is to improve the quantitative accuracy and to achieve cross-platform comparability of neat plasma proteome measurements.

Indeed, we are excited that we could show evidence for direct clinical utility of this SIL panel, without the need to expand the standard into a panel for disease-specific biomarkers, as we demonstrated for two diseases (COVID-19 and Loa Loa). However, this situation is rather a consequence of the selection of the analytically best peptides, rather than by design. Because we have focused on the identification of the most suitable peptides, we ended up covering a set of highly important and physiologically central plasma proteins, which are of medical, pharmacological, and diagnostic relevance. The proteins covered by the SIL panel include Apolipoproteins, proteins of the immune system, and coagulation factors, including known biomarkers such as C-reactive protein, Alpha-1-Acid Glycoproteins, Serum Amyloid A proteins, or complement factors C2, C3, and C9, to name a few. As such, the plasma proteome captured by the OSPP responds to a wide range of diseases and captures FDA-approved drug targets. We have reworked the language of our manuscript substantially to make this clearer.

Specific comments:

1. Page 4, Lines 14-28 and Page 22, Lines 5-15: Please recheck the mathematical annotations of the selection algorithm. Some elements are not well defined, and some expressions appear to be written in programmatic language (e.g. ==). It is not clear in which units were the ranks expressed and how were the relative ranks calculated. Was the $CV(p,n)$ calculated on log-transformed or raw values?

We thank the reviewer for the comment.

We have rechecked all the mathematical annotations for peptide selection in the manuscript. Every introduced variable is accompanied by its notation and description. Specifically, to avoid misunderstanding, functions like max or mean are supplied with an index indicating the variable over which the function is applied.

Re: programmatic language: The == refers to relational operators; as in the applied programming convention, one would distinguish between the assignment (“=”) and equality operator (“==”). Nonetheless, if this is confusing, we have replaced all equality operators “==” by an “=”.

We thank the reviewer for bringing up the point about the rank-based statistics. Here, we would like to point out that rank is unitless, and in our case, it changes from 1 (the lowest rank) to the number of precursors in a study ($N_p \sim 5E3$) (the highest rank). We have included the above information in our manuscript (Page 24,25, Methods).

As suggested we also provide more information about the calculations of the coefficient of variation (CVs.) Measurements in mass spectrometry are conducted on an intensity scale, and therefore the CV of precursor p in study n is calculated on an intensity scale. (Page 24,25)

2. It would be useful to the reader to specify whether during selection there was a condition that some project characteristics had to be considered or $PresInNProjects(p) \geq 4$ could mean that all four projects are EDTA-plasma. Would be useful to the reader to report which peptides were present in which project, to infer whether some are specific to a sample type. Apologies if I missed that information elsewhere, but I couldn't find it in Supplementary Table 1.

We thank the reviewer for raising this point. Yes, we had accounted for the possibility mentioned, and our selection strategy included the condition that a peptide must be present in more than 2 blood matrices. We apologize that this was hard to find and have now explicitly stated this criterion in the main text and Methods:

“We selected only proteotypic peptides quantified in at least half ($PresInNProjects(p) \geq 4$) of the examined studies and covering at least two blood matrices ($PresInNMatrices(p) \geq 2$).”

A peptide may be missing from the individual studies not only for the reason of its specific matrix and the MS setup applied but also for biological reasons. For this reason, we tested the detection of the selected peptides across the different matrices experimentally. In these experiments, 147 of the 211 peptides are detected in all four matrices, 174 in three, and 190 peptides were selected from studies in at least 2 matrices. This information has been included in Supplementary Table 1.

3. How did the sample type bias the ranks and relative ranks? Can this be estimated? This could be relevant for the coagulation factors or other proteins that are being captured in the coagulum. It would be relevant to explain this in the manuscript. It would be relevant to further explain the setting of blood sample collection and the processing that led to obtaining the plasma/serum samples.

We thank the reviewer for this comment. The reviewer raised three questions here:

- 1) How does sample/matrix type bias peptide ranks and relative ranks? Can they be estimated?
- 2) Are coagulation factors affected by matrix type?
- 3) How were blood samples collected and processed (since this may influence peptide detectability)?

Regarding question 1), yes, as expected, the matrix composition is an important factor for the peptide signals. In order to estimate this, we calculated rank variability using pairwise Wilcoxon signed-rank tests across blood matrices for each peptide. While a subset of peptides

exhibited stable ranks between matrices, others showed rank shifts between matrices ($p.adjust < 0.05$). We have included these results in Supplementary Table 1 and summarized the matrix test result in a PCA plot in Supplementary Figure 2. We have also added more detailed information on this matrix test in the manuscript:

“Quantification of each selected peptide, as expected, was also affected by matrix type (Supplementary Table 1). To assess peptide-specific abundance differences across matrices, we performed pairwise Wilcoxon signed-rank tests to evaluate rank shifts of the respective endogenous peptides between sample types. A subset of peptides maintained consistent relative abundance, with serum and heparin plasma showing the highest similarity—148 out of 188 peptides did not change rank significantly ($p.adjust > 0.05$). In contrast, substantial rank shifts were observed for the majority of endogenous peptides across other matrix comparisons (healthy cohort with 4 different matrices). The largest difference was seen between serum and citrate plasma, where 143 out of 194 peptides showed significant rank changes ($p.adjust < 0.05$) (Supplementary Table 1, Supplementary Figure 2b).”

Regarding question 2), our internal standard includes peptides from 7 out of 13 coagulation factors, including all three chains of Fibrinogen (Factor I), as detailed in the table below (adapted from Supplementary Table 1). While Factor IX (F9) was not detected in serum, at least one Fibrinogen chain was consistently observed across all evaluated matrix types.

Sequence	Genes	Pres_in N Projects	Pres in EDTA	Pres in Heparin	Pres in serum	Pres in Citrate
TGIVSGFGR	F10	5	1	1	1	1
VVGGGLVALR	F12	4	1	1	1	0
EQPPSLTR	F12	7	1	1	1	1
ELLESYIDGR	F2	8	1	1	1	1
VTGWGNLK	F2	4	1	1	1	0
SALVLQYLR	F9	5	1	1	0	1
GSESGIFTNTK	FGA	5	1	1	0	1
YQISVNK	FGB	4	1	1	1	0
YEASILTHDSSIR	FGG	4	1	1	0	0
VELEDWNGR	FGG	4	1	1	0	0

Table 1 in rebuttal letter, extracted from Supplementary Table 1 in the manuscript

The above table supports that for coagulation factors, there was no apparent bias against their selection. Furthermore, in our validation experiments, peptides from 5/7 included coagulation factors that could be reliably detected across all four matrix types (F2 and FGA in two matrices) (Supplementary Table 1).

Regarding question 3), Yes, absolutely, we agree that details on sample handling are important to interpret matrix-specific findings. We have now added a full description of blood

collection and processing procedures to the Methods-Blood Sample Acquisition. We also expanded the discussion, now explicitly spelling out the importance of standardizing sample collection and pre-analytics to achieve consistent results in plasma and serum proteomic studies.

4. How did the weighing approach bias the selection? E.g. if the maximum presence of a peptide was let's say 50% in a study and the same peptide had very low CV of 1%, the weight would have been the same as for a peptide that had 100% presence and 2% CV. Both CVs would have been low, but the latter peptide would have been detected in twice more samples. Would these two peptides be equally relevant for downstream selection, i.e. what is the rationale that the presence and CV should have equal importance in the weighing?

We thank the reviewer for this thoughtful observation.

In practice, peptides with high variability or low presence were deprioritized by our weighting scheme. This explains why we did end up with highly stable peptides in the final selection for the OSPP.

In more detail, while weight is calculated as the ratio of a precursor's (%) presence to the coefficient of variation (CV) ($\text{Weight} = \text{Presence} \times 1/\text{CV}$), they are not equally influential in the downstream selection. Peptides with low presence—even if they had excellent CV, they received lower weights and were filtered out in subsequent selection steps.

Specifically, peptides with a presence below 60% were not retained in the final standard selection, and only three peptides fall within the 60–70% range. Therefore, the scenario described by the reviewer (e.g., a peptide with 50% presence being weighted equally to one with 100% presence) did not occur in the selected set. Our rationale for this weighting approach was to ensure peptides were both consistently detectable and quantitatively robust while avoiding overly complex composite metrics. We have clarified this point in the manuscript.

5. Why was the lower cut-off of the relative rank average set to be 0.6?

We apologize for the typo. We actually used the threshold of 0.5. We have selected this threshold to select precursors that, on average, have a CV below the median CV. We added explanations into the Methods.

“Finally, the precursor's average (over considered studies) relative rank $\text{RelRank}(p)$ was used to select the best 'global' (i.e., non-project-specific) precursors for every protein, while we also required that the lower cutoff of the relative rank be set as 0.5.”

6. It would be important to specify in Figure 1 the stepwise exclusion of peptides at each step. The authors should specify the total number of peptides per project, followed by the number of peptides that are excluded at each step when a criterion is not met (e.g. how many peptides were excluded because they were not proteotypic or did not meet the chemical properties criteria, etc.)

We thank the reviewer for their comment.

We have included detailed selection information in Figure 1 (right panel) that gives the peptide numbers in each selection step and indicates the range in the text on Page 4:

“Number of precursors was changing from study to study, min = 4,653 and max = 6,361 with mean = 5,390 (Supplementary Table 1);”

“Eventually, this selection process identified 382 consistently quantified peptides.”

“Eventually, this selection procedure converged on 187 new peptides, expanded by 24 peptides that demonstrated excellent cross-platform performance as part of a previous SIL-IS panel designed to assess disease severity in COVID-19 and MPox patients.”

To avoid further overloading of the already complex Figure 1, we added more details (e.g., the size of the peptidome and how many precursors were proteotypic in each selected study cohort) in Supplementary Table 1 and Methods on Page 25 .

“The number of precursors was changing from study to study, with min = 4653 and max = 6361 (the full peptidome consisted of 10560 precursors, of which 7621 were proteotypic).”

“Eventually, this selection process identified 382 consistently quantified peptides. This is almost exactly 5% from the starting number of proteotypic precursors.”

We also indicated that the final set of peptides was about 5% from the starting peptidome.

7. Are some proteins represented by a single peptide? How has this affected the accuracy and precision in quantifying the different proteins? It would be useful to present the effect of the selection-related factors on precision in supplementary.

In our selected proteins, about 50% (71/211) are represented by a single internal standard. For the remaining 60 proteins captured by our peptide standard, 44 proteins are represented by 2 standard peptides, 13 proteins by 3 internal standard peptides, 2 proteins by 4 internal standard peptides, and 1 protein SERPINA3 by 5 internal standard peptides. We have added this detail in the main text:

“These 211 peptides are derived from 131 proteins, of which 71 proteins are represented by one internal standard peptide, 44 proteins are represented by 2 internal standard peptides, 13 proteins by 3 internal standard peptides, 2 proteins, APOA1 and APOB, represented by 4 internal standard peptides, and SERPINA3 is represented by 5 internal standard peptides (Supplementary Table 1, Supplementary Figure 1).”

We also added a barplot showing the distribution of the number of peptides per protein in Supplementary Figure 1.

While quantification is precise for the internal standards, one cannot generally estimate the precision at the protein level. The reason for this is that in the plasma proteome, proteins exist in multiple isoforms that share the same peptide; i.e., as in other SIL panels, the selected peptides are proteotypic but not isoform specific. However, because several of our proteins are represented by several internal standards, we indeed had the opportunity to get some estimates. We analyzed the correlation between peptides from the same protein model across different LC-MS setups on the example of the COVID-19 cohort. 40% of peptide pairs that come from the same protein model show a high correlation ($R^2 > 0.7$), but 60% of the peptides, despite being precisely quantified with the internal standard, show lower concordance, indicating a contribution of different isoforms to the peptide abundance.

Results : “Peptide-level analysis revealed that correlations between peptides mapping to the same protein showed diverging overall correlations; in each platform, over 40% of same-protein peptide pairs had an R^2 above 0.7 (Supplementary Table 7). It is worth speculating that the high isoform diversity in the human plasma is a main contributor to the diverging quantities of peptides derived from the same protein model.”

Discussion: “While OSPP-based internal standard workflows provide robust quantification at the peptide level, inferring accurate protein-level abundances remains a broader challenge in bottom-up proteomics, and specifically in plasma, which is rich in protein isoforms.”

8. Figure 2b – from where were the annotations obtained in the second, third and fourth panel from above?

We thank the reviewer for spotting this and apologize that we had not specified the source of these data.

The protein annotations for panels 1, 2, and 3 in Figure 2b were obtained and downloaded from the Human Protein Atlas (<https://www.proteinatlas.org/search>). We have added this information to the Figure legend.

The availability of clinical chemistry assays for the proteins covered by the OSPP has been surveyed by the Institute of Laboratory Medicine, Ludwig Maximilian University, Munich, Germany. We have acknowledged Prof. Dr. Daniel Teupser for this contribution in the respective section.

9. How was the limit of detection (LOD) determined? For proteins that have low abundance in non-diseased blood samples (such as CRP) – did the assay measure concentrations with the LOD below the reference values cut-off for that protein in a healthy population (where those are known) or was the LOD above the reference cut-off? This is important, since if the assay

is to be clinically relevant, it should be able to measure concentrations that would include reference values as well.

We thank the reviewer for the comment. LOD and LOQ were calculated based on replicates of the external calibration curve. LODs were determined using the standard formula $LOD=3.3\times\sigma/S$, where σ is the standard deviation of replicate y-intercepts of regression lines of the calibration curve, and S is the slope of the calibration curve. LOQs were determined based on triplicate injections of calibration standards, using a coefficient of variation (CV) threshold of ≤ 0.2 (or ≤ 0.4 if fewer than three points were available). We have to mention that both LOD and LOQs are instrument-related technical parameters (varying between instrument setup, LC-MS methods, and also the status of the instrument when we measure) and not OSPP-specific parameters.

Regarding the comparison to clinical reference values, our measurements focus on peptide-level concentrations rather than whole protein concentration and were performed in plasma rather than serum. Nevertheless, we compared our peptide LODs to reported CRP reference ranges in healthy adults. For example, among the three proteotypic peptides from CRP in OSPP, two are not detected in healthy control groups; one CRP peptide (RQDNEILIFWSK) is detected, and we calculated the LOD ranging from 2 to 12 pg/ μ L across different acquisition platforms. When we consider the dilution factor during digestion, the molecular weight of this peptide as $MW \approx 1764.925 \text{ g}\cdot\text{mol}^{-1}$, our LC-MS measurement, we would have 0.136-0.816 pmol of this peptide, and when we assume no loss during digestion and SPE, we would also have 0.136-0.816 pmol of CRP in plasma, which is roughly 3.4-20.4 ng of CRP in 5 μ l of Plasma, which LOD is roughly 0.68-4.08 mg/L similar to the reported healthy CRP values (~0.8-3 mg/L) (Delongui et al. 2013; Erlandsen & Randers 2000), demonstrating that our assay captures the physiologically relevant concentrations. Also here, we would like to highlight that both LOD and LOQ are dependent on the acquisition method used; it is not the property of the SIL-IS we introduced but rather a parameter of the analytical platform used.

We have updated the above information in Methods – Data Acquisition & Processing – Calibration Curve, and the calibration curve preparation is summarized in Supplementary Table 3.

10. Page 8, Lines 35-38: the COVID-19 cohort is not well balanced. It has a different number of patients with different levels of severity. I assume effective size is a typo referring to effect size. What does effect (or effective, if not a typo) size refer to in this sentence?

We thank the reviewer for the comment. The COVID-19 cohort includes the different treatment escalation levels, with reasonable case numbers, considering the size of the cohort. We apologize for having used the term, “Well balanced”, which indeed is subjective terminology. We would like to highlight that the cohort was not actively selected or stratified

based on disease severity. The only inclusion criterion was patient symptoms after a confirmed COVID-19 diagnosis in the early start of the pandemic in 2020; therefore, the distribution across severity levels (WHO 3 to WHO 7) occurred naturally among the population, and it is without intentional bias toward any particular group. We have removed the mention of “effective size” during the revision process in which the paragraph was extensively reworked. The revised paragraph for introducing the clinical cohort reads:

“To demonstrate the usability of our SIL-IS, we applied OSPP as internal standards in a clinical cohort study. We chose a well-characterized clinical cohort of severely ill COVID-19 patients (Kurth et al. 2020; Messner et al. 2021; Demichev et al. 2021; Wang et al. 2022) (n = 45, Figure 3a, Supplementary Table 4). This cohort, enrolled at Charité between March 1–26, 2020, included individuals across the WHO ordinal scale of treatment escalation, which serves as a proxy of disease severity (World Health Organization 2020). This reference cohort enrolled patients with WHO levels 0, 3, 4–5, and 6–7, ranging from uninfected to critically ill.”

We have also added an additional section in Methods—Clinical Study Design and Participants describing in detail how this cohort is selected:

“Patient samples were collected as part of an observational cohort (Demichev et al. 2021; Messner et al. 2020; Messner et al. 2021). The study protocol, including patient characteristics, treatment, and clinical outcomes, has been described previously (Kurth et al. 2020; Thibeault et al. 2021). Briefly, all hospitalized patients with PCR-confirmed SARS-CoV-2 infection treated at Charité – Universitätsmedizin Berlin, a tertiary care center, were eligible for inclusion regardless of age, gender, or disease severity, following written informed consent. To reduce selection bias toward less severe cases, patients requiring invasive ventilation were also included under a deferred consent procedure. The study size was determined by patient availability and logistical feasibility, and all enrolled patients were included in the analyses. Study date cutoffs were the 1st to 26th of March 2020. The study is registered in the German and the WHO international registry for clinical studies (DRKS00021688). The study was approved by the ethics committee of Charité – Universitätsmedizin Berlin (EA2/066/20). The cohorts are summarized in Supplementary Table 4.”

11. It is unclear how the design allows for “identification of peptides that are only abundant in either healthy or diseased samples, avoiding missing values due to low abundance”. Was there a pattern of missing values for some proteins that was related to the COVID-19 status? E.g. CRP levels would be elevated in a good portion of COVID-19 patients but not in some very mild cases or in healthy individuals. Shouldn’t the latter two groups have very low-abundant CRP levels that might be in some cases undetectable? Or was it undetectable regardless of the COVID-19 status? Same can be said for many of the other proteins for which there is strong evidence that they are elevated in COVID-19, like ORM1 and ORM2. The authors should explore if the missingness in the data is related to COVID-19 status.

We thank the reviewer for raising this point; we apologize if our approach in this regard was not explained in detail. It is expected that depending on the study and matrix and analytical setup, not all of the peptides (selected in a disease-agnostic way) are detected, and this could also be the case for specific disease groups. The way we dealt with missing peptide values in our COVID-19 use case is that we extracted the spectra of each peptide and checked detectability, applied a completeness filtering, and considered only peptides that were quantified in two-thirds of the samples. Additionally, we have reported the number of

samples where each peptide is above LOD in Supplementary Table 7. This approach ensures that peptides with excessive missingness—whether due to low abundance or technical issues—are excluded from downstream analysis. We examined the pattern of missing values across WHO severity grades and did not observe peptides that were consistently missing from any specific severity group (Supplementary Table 7 - Detection of peptide in certain Severity Group). This suggests that missingness was not systematically associated with COVID-19 status in our dataset but rather reflects the expected variability and lower abundance of these proteins in some healthy or mildly affected individuals. We have reported the detection of each peptide in each severity group in Supplementary Table 7.

We also checked our healthy cohort with different matrices; certain peptides/respective proteins show very low abundance in healthy individuals (Supplementary Table 1), which indicates the missingness is just due to low abundance of peptides not related to COVID status. We have clarified our filtering criteria as follows:

“Peptides that were detected in more than two-thirds of samples were included for subsequent analysis, and this criterion was met by 202 of the 211 peptides.”

12. There is a lack of clarity/consistency in the terminologies regarding the WHO severity scales for COVID-19. Seems that the WHO ordinal scale used in the study refers to the scale from reference 41 and not reference 42. The WHO ordinal scale has 8 orders (8 = death) – should be corrected in the manuscript. The reference 42 refers to the WHO clinical progression scale. In Figure 2, the authors write “WHO treatment escalation score”. It is important to clarify which scale exactly was used, to use the right terminology consistently, and to cite it properly.

We thank the reviewer for the comment, and we have revised the manuscript to achieve a fully consistent terminology throughout the manuscript.

We have cited the WHO treatment escalation score used to define disease severity, reference 41 (now 43) to support our approach to grouping WHO grades based on clinical progression. WHO grade 8 was omitted from our figures, as we have not collected and not analyzed any postmortem samples. We have also adjusted the Figure legends accordingly.

13. Although the focus of the study is on plasma proteomics method development, the cohort of COVID-19 patients needs further description – at least what is the distribution of age and sex because these factors can affect the plasma protein levels. How have age or sex biased the analysis of the association between protein levels and COVID-19 severity, since it is well known that male sex and older age are associated with severity and worse outcomes in COVID-19 patients?

We thank the reviewer for the suggestion.

In the revised manuscript, we have expanded the description of the COVID-19 cohort to clarify its composition and added in Methods a section of “Clinical Study Design and Participants”.

“Patient samples were collected as part of an observational cohort (Demichev et al. 2021; Messner et al. 2020; Messner et al. 2021). The study protocol, including patient characteristics, treatment, and clinical outcomes, has been described previously (Kurth et al. 2020; Thibeault et al. 2021). Briefly, all hospitalized patients with PCR-confirmed SARS-CoV-2 infection treated at Charité – Universitätsmedizin Berlin, a tertiary care center, were eligible for inclusion regardless of age, gender, or disease severity, following written informed consent.

To reduce selection bias toward less severe cases, patients requiring invasive ventilation were also included under a deferred consent procedure. The study size was determined by patient availability and logistical feasibility, and all enrolled patients were included in the analyses. Study date cutoffs were the 1st to 26th of March 2020. The study is registered in the German and the WHO international registry for clinical studies (DRKS00021688). The study was approved by the ethics committee of Charité - Universitätsmedizin Berlin (EA2/066/20). The cohorts are summarized in Supplementary Table 4.”

The reviewer asks reasonable questions about the role of age and sex in COVID-19 and their impact on the proteome. These questions have been extensively addressed in previous studies on this cohort (References 28, 33, 41, 42), and they confirmed that the observed proteomic differences are not primarily driven by demographic variables, including age or sex. We have revised our paper so that the previous work on the cohort is more visible.

“For example, the peptide quantities classified the patients according to the clinically observed disease severity according to clinical severity (WHO ordinal scale), with prior studies confirming these patterns are independent of age and sex (Kurth et al. 2020; Messner et al. 2021; Demichev et al. 2021; Wang et al. 2022)”

14. Figure 3b: PCA is not clustering. What do the ellipses represent? Information on these analyses are missing in the methods section too.

We thank the reviewer for pointing this out.

We had included confidence ellipses at the 80% level as a tool to visualize the within-group variance and the distribution of samples around the group centroid. We apologize if this may have been confusing, and we noticed the ellipse is not needed in this plot; therefore, we have removed the ellipse for better visualization and revised the legend of Figure 3b. We have also refined the terminology in the main text and added details in the methods about this analysis.

15. Page 11, Line 34: I assume the authors refer to Figure 3c. Why were the changes in protein levels normalised to WHO grade 3 patients? Wouldn't it be a more intuitive comparison to normalise the fold changes to healthy controls?

We thank the reviewer for pointing out the typo. We were indeed referring to Fig 3c and have corrected it. In this figure, we chose to normalize fold changes to the WHO grade 3 (mild COVID-19) group primarily for illustration purposes and to ensure consistency with our

previous work (Wang et al. 2022). We have now clarified this rationale in the figure legend and main text

16. Figures 3c and 3d nicely present a gradual change related to COVID-19 severity. However, the authors should further explain why C3 and CST3 are presented in Figure 3d, since the published evidence for their elevated levels in COVID-19 patients is variable. Wouldn't it be more relevant to prioritise other proteins for presentation? The authors should include more references related to the claims on Page 11, Lines 34-45.

We thank the reviewer for the comment. C3 and CST3 might not have received much attention in the context of severe COVID; however, we have detected an increase in C3 and CST3 in severe COVID also previously (Wang et al. 2022) and by being connected to the complement to kidney function, they are part of biological processes of direct relevance to severe COVID-19 disease. We thus concluded that highlighting understudied rather than well-known COVID-19 proteins gives extra value to our study.

Following the reviewer's suggestion, we have added more references to our previous work, as well as relevant studies by others, to support that the peptides discussed behave consistently with our prior observations. The text and additional references have been updated accordingly to reflect this alignment.

17. Supplementary Table 5. Would it be valid to use the Kendall's Tau trend test for an analysis of trend where the orders are not proportional? In the analyses, how were the patient groups treated numerically – as orders of 0, 3, 4, 5, 6, and 7, or from 0 to 5, or as grouped based on the severity group as 0 (healthy), 1 (mild), 2 (moderate), and 3 (critical)? The authors should explain the rationale for their analytical decision and how a different categorisation of the order could have affected the analysis.

We thank the reviewer for asking about the statistics. By being a non-parametric test that measures monotonic association between two variables Kendall's Tau (following WHO groups 0, 3, 4, 5, 6, and 7) is applicable to our study. We consider it suitable in this context, as it works on ranks rather than assumptions of equal group sizes or distributions. In order to give additional confidence to the differences of the severity groups and match what was shown in boxplots, we added the statistics using ordinary logistic regression (following severity: healthy, mild, moderate, and critical); we did not see an obvious change in whether the peptide is statistically significant in COVID-19.

To respond to the reviewer, we included in the statistics both Kendall's Tau trend test (based on different WHO grades) and ordinal logistic regression (based on severity groups) and updated the output in Supplementary Tables 5, 6, and 7 and the respective statistical approach in the Methods; in Figures with boxplots, we have updated the p-value using the statistical method that matches the plot.

18. The boxplots show data related to the groups described in Figure 3a. They should be presented in the same manner the orders are defined in the trend analysis if the authors justify the validity of the method. It can be helpful to the reader to add p values for the comparison in the boxplots. How have age and sex affected the association between the plasma proteins and COVID-19? The protein and gene names that correspond to the peptides should be included in the supplementary tables in all sheets that show peptide information.

We thank the reviewer for this observation.

We agree that our previous statistics of Kendall's Tau trend test follow WHO groups (WHO 0, 3, 4, 5, 6, and 7). Although a valid statistical method, the box plots are shown based on severity groupings, which caused some confusion. We have now changed the adjusted p-values derived from the ordinal logistic regression analysis (which is calculated based on severity group to match what the boxplot shows) for boxplots in Figure 3d (as well as Figures 4g, 5d, and 5e) and mentioned the statistical method in the figure legend.

We acknowledge that age and sex influence plasma protein profiles in COVID-19; however, as this manuscript focuses on the development and application of the OSPP internal standard, and our previous studies have addressed the clinical aspects in detail, we chose not to expand on this topic here to maintain a clear focus on OSPP.

Supplementary Table 1 has summarized the peptide sequences along with their corresponding gene and protein names. Since all quantification was performed at the peptide level, to avoid redundancy and to avoid causing confusion, we did not repeat protein names in supplementary tables where protein-level information was not required/reported.

19. How consistent is the association between A2M and COVID-19 based on published evidence?

A2M was also detected as a decline in severe COVID-19 in previous works (REF 49,50). We have added this information to the manuscript.

20. Page 16, Lines 35-40: The % related to the variance explained differs from the one shown in Figure 5c. Please recheck. In the format it is presented, it is not clear from the PCA that the primary source of variance is related to biology.

We thank the reviewer for pointing this out. We have corrected the numerical inconsistency in the text.

For easier visual interpretation, we have split figure 5c into 2 subplots. The biplot on the left panel of Figure 5c in the revised manuscript indicates that the variation across disease severity groups aligns with PC1, while platform-related differences align more with PC2. Since PC1 explains a larger proportion of variance, this indicates that biological variation contributes more than technical variation.

To further support this, we added variance partitioning analysis across peptides. When normalized by OSPP ratios, biological factors explained a median of 15.5% of variance compared to 5.1% from the LC-MS platform. In contrast, using endogenous peptide normalization (norm_light), biological variance was only 8.5%, while platform effects explain 39.6%. These results confirm that OSPP normalization enhances biological signal over technical variation.

We have added the above analysis output to Supplementary Table 7 - variance partitioning across peptides—and updated the manuscript and details of analysis in Methods. Now the manuscript reads:

“Variance partitioning analysis across peptides further confirmed the dominance of the biological signal versus technical differences between the platforms (Supplementary Table 7). Among all peptides, the treatment escalation level explained a median of 15.47% of the variance, while the technical differences between the acquisition platforms caused only 5.07% of the variance.”

21. Page 17: the use of the term “composite cohort” is misleading. Based on the information presented in the manuscript, the cohort is not a separate cohort nor a separate MS run but a computationally generated dataset for the purpose of comparing the agreement in quantifications of the OSPP between a single platform and a hypothetical scenario where the sample could have been analyzed on any of the platforms. If that is correct, the authors should explain this more clearly and avoid expressions like “two cohorts”. The authors should also compare the agreement between each of the single platforms and the agreement between the randomly sampled dataset and the remaining platforms to demonstrate that the high agreement is reproducible when choosing any of the other single platform runs. When generating the randomly sampled dataset, the authors should consider excluding the single platform they intend to compare the randomly sampled dataset to, since sampling from the dataset of the comparing platform would imply a leakage of information, overestimating the agreement.

We thank the reviewer for the helpful clarification. We apologize that the term “composite cohort” was misleading and have revised it to “composite dataset” and ‘artificial dataset’ throughout the manuscript to reflect that this is a generated dataset that computationally combines randomly selected data from existing datasets, not a separate biological cohort or MS run. We have revised the main text:

“To further highlight the possibility for cross-platform data alignment using the OSPP internal standard, we evaluated whether data collected with different acquisition platforms can be mixed to produce comparable

results to those from a single platform. We created an artificial "composite dataset" with peptide quantities randomly taken from datasets acquired from various platforms (OSPP normalized), reflecting the sample and severity distribution of the original clinical cohort (Supplementary Table 7)."

Regarding the reviewer's suggestion for additional comparisons between single platforms and randomly sampled datasets, we fully agree that such an analysis would provide a more comprehensive assessment. However, conducting all possible randomizations and cross-platform comparisons would represent a very large and complex study beyond the scope of the present work. Instead, we designed the composite dataset comparison to illustrate one possible and practical approach, which we believe the current manuscript sufficiently demonstrates the reproducibility and agreement across platforms.

To address the concern about information leakage, we have now updated our resampling procedure to exclude data from the comparison platform (e.g., hZsSWATH) when generating the composite dataset. This avoids circularity and ensures an unbiased comparison. The respective statistical analyses are updated in Supplementary Table 7.

22-1.. Page 17, Lines 19-25: The Bland-Altman plot analysis is unclear. Is the quantification consistency measured in two cohorts or in two different runs on the same samples from the same cohort on another platform (see comment above)?

The quantification consistency is measured for each peptide in two cohorts; one is from a dataset generated from a single platform, and the other is the composite dataset. In order to improve clarity, we have revised both the manuscript and the figure legend and added in detail of the Bland-Altman analysis in the methods. We revised the text; it now reads:

"To assess quantification consistency between the composite dataset and individual dataset, we performed Bland-Altman analysis for each peptide, using all corresponding sample measurements from both datasets (Supplementary Table 7)."

22-2. Figure 5f is presented in relation to the claim that the quantification between the two measurements agree nicely but it represents a single peptide (from A2M) and not the agreement in all peptides.

We thank the reviewer for this valuable comment.

It is correct that Figure 5f displays a Bland-Altman plot for a single peptide (from A2M). Applying Bland-Altman analysis to multiple peptides simultaneously would not be correct, as it would mix distributions from different peptides. To address the suggestion of the reviewer to show more peptides, we have reported the low median bias and low limits-of-agreement range across all peptides in the main text and provided an overall measure of agreement between the two datasets. Now the text reads:

“Bland–Altman analysis across all peptides that were above the limit of quantification showed a median absolute bias of 0.037 between datasets, with an average limits-of-agreement range of 1.716 units (Supplementary Table 7). Notably, 76.1% of peptides demonstrated differences within $\pm 10\%$, indicating high concordance between quantification methods and 96.5% of the peptides showing bias between -1 and 1, signifying significant concordance in peptide ratios between the two datasets (Supplementary Table 7);”

Additionally, for the remaining peptides that did not show bland-altman results as a plot, we have included the individual Bland-Altman statistics for each peptide in Supplementary Table 7 - bland-altman output.

22-3. Lines 23-25: This segment refers to the agreement in quantifications of all the peptides and should be presented as a main figure instead. The bias close to absolute values of a difference of 1 in some proteins is not negligible.

We thank the reviewer for the suggestion. Referring to the comment above, we have listed the detailed information of all Bland-Altman statistics in Supplementary table 7/-sheet “bland-altman” and report the median of bias in the main text.

For the analysis, we calculated the mean of the absolute bias (using non–log-transformed ratios) and agree that several peptides exhibit considerable bias. We acknowledge that proteome digests are highly complex, and in high-throughput analyses certain limitations remain, particularly for low-abundance peptides or under strong matrix effects. Our focused analysis of peptides with absolute bias >1 suggests that discrepancies largely arise from differences in instrument sensitivity, which can push signals close to the limits of quantification, as well as variations in ionization across platforms, e.g. peptide, EGPYSISVLYGDEEVPRSPFK, from protein FLNA. We want to mention that these peptides represent only a minority and do not significantly affect the overall agreement, as demonstrated by the near-zero median bias across all peptides. We keep those peptides with absolute bias > 1 in the OSPP, as they may prove useful when detected with more sensitive methods or at elevated levels.

23. The claim on page 19, Lines 29-31 is not supported by the presented evidence. The OSPP is not validated for patient stratification and outcome prediction. The authors have demonstrated that some proteins have a gradual increase or decrease in levels at the time of a given COVID-19 severity. OSPP is not built into a model that provides patient stratification and outcome prediction.

We apologize if our wording caused confusion. Certainly, OSPP is a disease-agnostic internal standard for MS proteomic experiments, and we have not conducted prospective clinical studies using it as a marker panel herein. Having said that, we conducted such studies in the past using a biomarker panel, with several peptides also contained in the OSPP (Wang et al.

2022). These studies confirmed the classification potential of highly abundant plasma proteins, including those captured by the OSPP, in acute COVID-19. We have revised the wording:

"For example, the peptide quantities classified the patients according to the clinically observed disease severity (WHO ordinal scale), with prior studies confirming these patterns are independent of age and sex (Kurth et al. 2020; Messner et al. 2021; Demichev et al. 2021; Wang et al. 2022)"

24. Were the DIA data log-transformed prior to normalisation?

The data shown has not been log transformed prior to normalization. We have clarified this in the Methods - Data analysis:

"All quantitative data presented and used for statistical analysis are in linear scale and were not log-transformed at any stage of the analysis."

25. Why was the CV calculated based on median and median absolute deviation and not mean and standard deviation? Did this approach underestimate the CVs?

We thank the reviewer for raising this question regarding CV. We chose to calculate CVs using the median and median absolute deviation because proteomics datasets frequently contain outliers and show skewed distributions, where mean and standard deviation can give biased estimates. The median-based approach provides a more robust measure of variability under these conditions. We agree, however, that CVs calculated this way can sometimes underestimate variability. To account for this, we reported both robust ($CV = MAD/median$) and conventional ($CV = SD/mean$) CV values, ensuring transparency and enabling direct comparison. The additional data are provided in Supplementary Tables 6 & 7.

26. Some analyses were performed but were not described under statistical analysis and visualisation.

We thank the Reviewer for the suggestion. In response, we have revised the Methods – Data Analysis – Statistical Analysis section (Page 34) to provide a clearer and more detailed description of our statistics. Specifically, we now include the exact R package versions and functions used for statistical tests (e.g., for linear modeling, variance partitioning, and differential expression analysis) and the statistical formulas or model structures where applicable.

27. Some examples of typos that the authors should correct are provided below. There were more – the authors should carefully recheck the manuscript.

- Page 3, Lines 31-33: maybe these two sentences were supposed to be connected by a comma?
- Page 4, Line 7: there is an additional “from” in front of prepared.
- Page 8, Line 1: “in combination” with is missing?
- Page 8, Line 20: sentence unclear.
- Page 11, Line 30: a closing bracket is missing. And the subsequent sentence is unclear and long.
- Discussion: the second sentence is long and unclear, with some typos.

We thank the reviewer for spending time checking the language. We have gone through the whole manuscript and corrected accordingly.

Reviewer #2 (Remarks to the Author)

The manuscript by Ralser and coworkers presents the development, validation, and application of the Charité Open Peptide Standard for Plasma Proteomics (OSPP). This is an open-source and cost-effective stable isotope-labeled internal standard designed to improve precision, reproducibility, and cross-platform comparability in plasma proteomics workflows. The peptide selection process is methodical, combining empirical data from thousands of plasma proteomes and control injections. The use of a relative rank metric ensures the selection of analytically robust peptides with low technical variability. The validation of the OSPP across different platforms (e.g., ZenoTOF 7600, timsTOF, and Orbitrap Exploris 480) demonstrates its versatility and robustness. The authors demonstrate the utility of OSPP for both targeted and untargeted proteomics across multiple LC-MS platforms in a case study involving a COVID-19 patient cohort. The manuscript is easily readable and highlights an affordable technique that could help standardize plasma proteomics. However, I have a number of comments and concerns that need to be addressed before I can recommend publication.

Specific comments to address:

1. I assume that all the proteotypic peptides selected for the OSPP standard are tryptic peptides but this is not specified in the manuscript text. Were the selected peptides derived from other proteases than trypsin or are they all fully tryptic peptide sequences? This should be clearly explained in the manuscript text.

We thank the reviewer for the comment. The currently selected peptides contain only fully tryptic peptides that are conservatively estimated to be proteotypic. We have clarified it in the main text:

“To prioritize the most reliably quantified precursors from tryptic peptides that are proteotypic to the respective protein, we introduced a relative rank metric.”

2. The peptide/protein relationship is an issue that should be more directly addressed. The authors are very open that they are only evaluating peptide quantification but the abstract and the title reflects that they are also looking at protein quantification even though this is not assessed in the manuscript. This should be addressed. Preferably the potential on protein level quantification should be evaluated and if not the abstract should reflect this and the text

should discuss the connection. Orthogonal measurements (eg. ELISA or DIA) of at least some proteins would also be beneficial.

We appreciate the reviewer's thoughtful and constructive comment. We acknowledge the problem of estimating accurate protein levels from peptide levels, but this is a challenge not specific to the OSPP but a general research subject in the bottom-up proteomics field. The problem has indeed a specific nuance in human plasma, which is rich in protein isoforms; similar to other SIL-panels, our peptides are proteotypic, but they are not isoform specific. Our internal estimates are that each plasma protein exists on average in 10 isoforms, but the community still lacks a reliable ground truth about this (see also our reply to Reviewer #1, comment 7). Acknowledging the reviewer's point, we have thus included a paragraph about this in the discussion. It reads:

"While OSPP-based internal standard workflows provide robust quantification at the peptide level, inferring accurate protein-level abundances remains a broader challenge in bottom-up proteomics, and specifically in plasma, which is rich in protein isoforms. Thus, while peptide-centric quantification remains robust and reproducible, reliable protein-level estimation, especially in complex matrices like plasma, remains non-trivial. Computational methods such as MaxLFQ (Cox et al. 2014) have improved protein inference from peptide intensities, yet the complexity of the proteoform space creates limitations, not the least of which is that some information is inadvertently lost by digesting the intact proteins into peptides. In this study, we chose not to further address the pre-analytical factors, such as blood handling and processing, which remain a key challenge in plasma proteomics. Nonetheless, the use of an internal standard like OSPP can help disentangle technical from pre-analytical variation by providing stable reference signals across samples. This capability is important to extend OSPP's utility as an internal standard for reliable application in diverse clinical proteomics contexts and multi-laboratory clinical studies."

3. While the OSPP includes 211 peptides from 131 plasma proteins, this represents only a small subset of the plasma proteome. Key biomarkers or protein isoforms relevant to specific diseases may be missing, limiting its applicability in certain contexts. To demonstrate the advantages and also the limitations of the OSPP standard, the authors should benchmark the performance of the OSPP standard against the state-of-the-art peptide standards for plasma proteomics, i.e. the PQ500 standard from Biognosys. This benchmark should be performed evaluating plasma proteomics from a clinical cohort in terms of biomarker coverage and peptide and protein quantitation.

The OSPP internal standard has been selected from a series of untargeted neat plasma proteome datasets, which all focused on the high abundant plasma proteome. It is correct that the OSPP therefore does not contain standard peptides for the quantification of lower abundant proteins. As suggested by the reviewer, we have added a benchmark against the commercial PQ500 reference standard in our revision.

The benchmark was possible because 81 peptides overlap between both peptide standards. We have tested whether the endogenous peptides that are specific to one of the standards have better or worse quantification characteristics. For this we used two datasets. First, we performed triplicate injection of OSPP spiked into a pool of EDTA plasma and analyzed in micro-flow LC coupled to a ZenoTOF. Second, we used a DIA dataset including 10 controls and 10 colorectal cancer samples from PRIDE (PXD036594) (Lesur et al. 2023) with a longer gradient (100 min).

The repeated injection showed that

- Detection consistency: Although PQ500 contains 804 peptides from 572 proteins, many endogenous counterparts were not detected consistently in this setup (43% consistently identified, 352 peptides from 213 proteins). In contrast, the OSPP, 207 of the 211 peptides derived from 128 proteins were constantly quantified (Supplementary Table 1)
- The 81 shared peptides (endogenous) showed consistent quantification performance (median cv = 5.45%). For proteins represented in both panels but with different peptide selections, shared protein OSPP-specific peptides (n = 108) showed a median CV of 6.12%, while shared protein PQ500-specific peptides (n = 133) had a higher median CV of 7.31%.
- The overall reproducibility of OSPP shows similar reproducibility (median cv = 6.11%) and was more precisely quantified compared to the overall PQ500 peptides (median cv = 7.96%) (Supplementary Table 1 - CV_compare with PQ500_peptide, Supplementary Figure 3a).

PQ500 includes peptides for lower abundant proteins that can usually not be measured in high-throughput plasma proteomics but need more sensitive LC-MS methods. Therefore, we also reprocessed a dataset acquired from top14 protein depleted samples on a nano-flow LC-MS setup with a 105 min long gradient which included PQ500 SIL-IS..

Approximately, 60% of PQ500 peptides (489/804) were consistently identified with median CVs of 32% (CRC) and 27% (controls). In contrast, the peptides selected for the OSPP showed similar performance: > 90% coverage for 194 peptides from 124 proteins, with median CVs of 31% (CRC) and 25% (controls).

All data mentioned above have been included in Supplementary Table 1 and Supplementary Figure 3.

The above internal and external datasets all shows that while the PQ500 contains more peptides overall, our selected OSPP peptides were more consistently detected and more precisely quantified. Additionally, in the clinical use case we presented in manuscript, 187 pairs (88.6%) of endogenous and its matched SIL-IS OSPP peptides were consistently

identified and quantified across more than two-thirds of the samples, across three LC-MS platforms and five different acquisition and quantification methods, which further indicate the selection of OSPP peptides is applicable to various LC-MS setups.

4. It is great that the authors evaluated different plasma/serum material and different MS setups. Additionally different protein digestion protocols or different centers would also be very relevant to test the true potential for standardizing. It should at least be discussed as a limitation.

We appreciate the reviewer's insightful comment, and we could not agree more. Sample handling and pre-analytics are a crucial factor for the cross-study comparability of blood proteome studies, and for converting the panel into an analytical test, several additional steps, including multi-center studies, will have to be conducted. However, given the complexity and the large space of possibilities of how serum and plasma can be collected, stored, and processed, evaluating such endeavors would exceed the focus of this study.

Notably, the OSPP is indeed part of an ongoing multi-center study using different digestion protocols and LC-MS setups; this would provide us a better understanding of true potential and standardization using the SIL-IS. These results will be presented as part of a future study.

5. The OSPP was primarily validated in the context of a COVID-19 cohort. While this is a relevant use case, additional studies in other diseases (e.g., cancer, cardiovascular diseases) would strengthen the generalizability of the standard for broader clinical applications.

We thank the reviewer for this thoughtful comment. We have applied the OSPP as an internal standard to study a neglected tropical disease, loiasis, and have published the results in parallel (Dierks et al. 2025). Loiasis is a filarial parasitic infection endemic to parts of Africa—an application context that is clearly distinct from COVID-19. We have added a discussion of these points in the revised manuscript:

“For example, we applied the OSPP in a well-characterized COVID-19 cohort (Kurth et al. 2020; Messner et al. 2021; Demichev et al. 2021; Wang et al. 2022), where we detected many disease-responsive proteins. In a parallel study, we applied the OSPP in the characterization of Lasis (Loa Loa), a rare and neglected tropical disease characterized by an eyeworm infection. OSPP facilitated identification of potential biomarkers and classifiers of disease stage (Dierks et al. 2025).”

“Thus, by adding additional peptides or by recombining individual peptides into new panels, the OSPP can be converted into both smaller disease- or indication-specific marker panels or expanded to capture a broader range of the proteome.”

6. Page 4, line 17: It is a good strategy to select peptides based on previous knowledge of detection and reliable quantification with LC-MS. The strategy will be biased by the samples included in the datasets. And if the criteria deselect peptides with low CVs in a database using non-healthy and healthy samples, then peptides with biological dynamics in disease states could be filtered out even though they might be well quantified. Similarly, the resulting list includes several IgG-related genes that might be of questionable value from a biomarker perspective. Is there a rationale for including so many IgG peptides?

We thank the reviewer for this thoughtful observation. To minimize the risk of excluding peptides with disease-related biological dynamics, we based our peptide selection on repeated measures from study pools rather than from individual healthy or diseased samples. This approach prioritized technical reproducibility while avoiding direct bias toward any clinical group. We have clarified in the revised manuscript in methods:

“These samples were pools prepared from human plasma and serum using a semi-automated workflow in 96-well plates, which involves a clean-up step using solid-phase extraction before being analyzed on a proteomic platform that uses analytical flow rate reverse-phase chromatography with water-to-acetonitrile gradients and a throughput of 5-8 minutes/sample (Messner et al. 2021). Proteomes were recorded using data-independent acquisition on two SCIEX TripleTOF 6600+ instruments operating in Scanning SWATH (Messner et al. 2021) mode.”

Immunoglobulin-derived peptides were not excluded on purpose, as the selection process was untargeted, disease agnostic, and strictly driven by technical performance metrics (e.g., reproducibility, detection rate, signal quality). While we agree that IgG peptides can be difficult to interpret biologically, they have shown value in predictive modeling and patient stratification in our previous studies and are thus valuable to be quantified as part of untargeted proteomic studies (references 28, 33, 40, 41).

We have highlighted in the Results section that peptide inclusion in the OSPP standard as an internal standard was based on technical robustness across diverse datasets, not on biological specificity. To address the reviewer’s concern, we have also expanded the Discussion to highlight the modular and adaptable nature of the OSPP internal standard—allowing users to customize the selection for disease-specific peptides as needed. Now the revised discussion reads:

“In a parallel study, we applied the OSPP in the characterization of Lasis (Loa Loa), a rare and neglected tropical disease characterized by an eyeworm infection. Here the application of the standard allowed to propose potential biomarkers and classifiers of disease stage (Dierks et al. 2025)”

7. Page 8, line 1-14: The robustness test across different plasma/serum material is very relevant. There is no figure to illustrate these data. Should be included – at least supplementary. Information regarding the material handling (centrifugation steps) is lacking.

We thank the reviewer for the helpful suggestion. We added additional information related to different matrix studies in Supplementary Table 1 and Supplementary Figure 2.

In response to the reviewer's comment, we have added the material handling procedure (Blood sample acquisition) in the Methods section. In short, all blood samples were centrifuged at 2000 G, 4°C for 15 mins and then frozen at -80°C until further sample preparation and analysis.

8. Page 14, line 12: Why is A2M chosen as example? It would be nice to see examples of differential abundance. E.g. a peptides from each "concentration bin".

We thank the reviewer for the suggestion. We choose A2M (peptide FEVQVTVPK from concentration bin 3) because of its strong responsiveness to COVID-19 severity, in this and in previous studies (REF 49,50). We also use A2M as an example throughout the manuscript to be consistent.

As suggested by the reviewer, we have selected additional examples from the different concentration bins, namely

peptide ALDFAVGEYNK from CST3 (from concentration bin 1, 10 pg/μl in OSPP);

peptide EITALAPSTMK from ACTA (from concentration bin 2, 50 pg/μl in OSPP);

peptide LLDNWDSVTSTFSK from APOA1 (from concentration bin 4, 2 ng/μl in OSPP).

The boxplot for peptides from different "concentration bins" shows that SIS peptides could help provide robust quantification despite the different concentrations in OSPP. We have added the respective figure as Supplementary Figure 3.

9. Page 15, figure 5c: The PCA plot is central to the result section, and I do not think the technical vs. biological accuracy is very clearly shown.

We thank the reviewer for pointing this out. In response, we have revised Figure 5c into two subplots for better visualization. The biplot on the left panel integrates both the PCA scores and loading vectors to provide a clearer visualization of the relationship between peptides, clinical groups, and acquisition platforms. Additionally, we have added on the right panel of Figure 5c the same PCA scores along with 80% confidence ellipses for each group to improve clarity in the separation and variance explained by the principal components. The variation across disease severity groups aligns with PC1, while platform-related differences are much less pronounced and align more with PC2.

To further support this, we performed an additional analysis: variance partitioning across peptides. When normalized by OSPP ratios, biological factors explained a median of 15.5% of

variance, compared to 5.1% from the LC-MS platform. We hope this could better explain that the technical variation is lower than the biological variation.

These results confirm that OSPP normalization enhances biological signal over technical variation.

We have added these analyses to Supplementary Table 7 and updated the figure, legend, manuscript, and methodological details.

“Variance partitioning analysis across peptides further confirmed the dominance of the biological signal versus technical differences between the platforms (Supplementary Table 7). Among all peptides, the treatment escalation level explained a median of 15.47% of the variance, while the technical differences between the acquisition platforms caused only 5.07% of the variance.”

10. Page 19, line 4: “Indeed, the use of an internal standard can also be helpful to improve the quantification of peptides that are not covered by the standard”. This is potentially a strength of the standard but is not evaluated in the text nor is there any references to back up the claim. The text should reflect this or find references.

We thank the reviewer for pointing this out. Indeed, this is an outlook inspired by metabolomics studies of others (Ulvik et al. 2021; Bruheim et al. 2013) as well as proteomics studies on microbiome (Picotti et al. 2009). We indeed believe there is potential in plasma proteomics to use internal standards to normalize other physio-chemically similar peptides, but thus far, the field lacks appropriate computational methods to effectively make good use of this. We have thus also expanded the limitation section accordingly.

“Moreover, works in metabolomics and microbiome proteomics have shown that SIS are not limited to improving the quantification, or to estimating absolute concentrations, of the directly matched, endogenous molecules but can potentially also be used as surrogate internal standards for a much broader set of peptides and proteins (Ulvik et al. 2021; Bruheim et al. 2013; Picotti et al. 2009). We anticipate that future algorithmic developments could unlock this potential also for proteomics, opening new opportunities for broader quantitative harmonization.”

11. Page 19, line 8: “simplifying data analysis is an underestimated benefit of SIS panels”. It does not appear that the overall data analysis will be more simplified with SIS. It requires specific software and there are problems regarding peptide detection etc., so it could in some cases also make it more complex.

We thank the reviewer for raising this important question regarding data analysis.

We agree that one needs to distinguish the raw data processing and the follow-up data analysis here and apologize that we had not made this distinction sufficiently clear. Indeed, we could not agree more that the raw data processing part becomes more complex when

using internal standards, and current software solutions do present challenges in this context. To address this, we have detailed our approach to generating the spectral library and provided the data processing pipeline specifically developed for analyzing samples containing OSPP (Methods—Generation of OSPP-specific Human Spectral Library, Raw Data Processing).

However, the situation is different at the downstream, post-data processing stages; specifically in large clinical studies, we usually observe huge challenges in post-acquisition steps—particularly in identifying and correcting batch effects and fulfilling the regular task to compare results across studies. In this regard, internal standards like OSPP can significantly streamline and improve the robustness of downstream analyses. We have updated the manuscript to clearly distinguish between these two aspects and hope it's clearer now:

“A future challenge involves enabling hybrid analyses that simultaneously perform absolute quantification using internal standards and discovery-based proteomics within a single DIA experiment. Although current software tools such as DIA-NN (Demichev et al. 2019) or Spectronaut (Bernhardt et al. 2012) partially support this, these functionalities could still be optimized, as they were not originally designed specifically for this purpose. We provide a protocol for generating an OSPP-inclusive spectral library along with a corresponding data processing pipeline optimized for DIA-NN. While the inclusion of SIL-IS peptides increases data processing complexity, many bioinformatic processing steps that are required further downstream profit from the presence of an internal standard and can be simplified. For example, because SIL-IS peptides provide internal correction for each sample, intensity distributions do not require alignment, and biases such as instrument performance variation and time-dependent drift are inherently corrected. This situation greatly simplifies the integration and joint analysis of samples from different batches, instruments, or sites.”

Reviewer #3 (Remarks to the Author):

This paper logically lays out the selection of 211 tryptic peptides that arise from plasma proteins and the development of these peptides into a standard “kit” that can be used across both targeted and untargeted (DIA) mass spectrometry-based plasma studies. The authors provide an example of its application in a small COVID-19 patient study. Finally, they provide the information needed for the inclusion of all or some of these peptides in the spectral libraries that would be needed from transferability.

This is a very well-designed approach, and it has been exceptionally well-executed. The ability to have a set of SIL peptides in which the performance has been well-characterized and that reach the preclinical requirements will allow, if adopted, cross-comparison between mass spectrometry platforms and workflows, different laboratories and, importantly, act to allow normalization between cohorts.

Important considerations.

1. It would be very helpful if the authors included a dynamic range figure that shows the concentration range of each endogenous peptide and the concentration of the corresponding SIL peptide.

We thank the reviewer for their positive overall assessment and this suggestion. We have generated a dynamic range plot and included it in the manuscript as Figure 2d on Page 7.

Figure 2d: Dynamic range and concentration matching of OSPP peptides to endogenous plasma levels. The dynamic concentration range of OSPP peptides is based on calibration curve-derived concentrations measured in pooled EDTA plasma using microliter flow-rate chromatography and Zeno SWATH DIA on a ZenoTOF 7600 instrument (SCIEX). Peptide concentrations were estimated by comparing endogenous peptide signals to a measured calibration curve. A ratio (endogenous/heavy) difference within one order of magnitude indicates successful concentration matching.

2. The authors need to include the retention times and the retention time variability for each peptide, at least across the Covid study. If possible, also in the study using the Exploris 480 instead of the ZenoTOF 7600. This is important as this will impact whether these peptides can

be used as retention time standards (replace IRT in DIA studies) as well as quantitative internal standards.

We thank the reviewer for the valuable suggestion.

We have included the RT information for both endogenous and standards in reference 52 - DIA-NN output reports that report retention time for each peptide in each platform—and Supplementary Table 7, the differences between OSPP and endogenous and the range and CV of the retention time of each precursor in each platform.

Indeed the peptides show stable retention times throughout measurement and are also well-distributed along gradients; thus, they can be used as RT standards. We have evaluated the retention times of each peptide across all DIA platforms using the measurements conducted for the COVID cohort. Now the statement reads:

“Retention times (RTs) were consistent between OSPP and respective endogenous peptides, with a median difference less than 200 ms, indicating that the peptides were correctly identified (Supplementary Table 7). Moreover, the RTs for each peptide remained stable across all samples, with RT fluctuations across all samples within the acquisition batch below 40 seconds and a median CV less than 2% in each platform (Supplementary Table 7).”

3. The authors discuss potential expansion of the multiplex but also need to state (speculate) the minimum number of peptides needed for the SIL reproducible/cross valuation kit. This can be part of a “next stage/limitation section,” which could include deployment across more instruments and laboratories. The authors may also want to address how this could be used for comparison to other proteomics studies that measure protein/proteoform using capture (or other) reagents.

We thank the reviewer for raising this point.

To our knowledge, there is no strict minimum number of peptides required for an internal standard—historically, even single peptides have been quantified. Having said that, the strength of targeted proteomic approaches/peptide biomarker standards lies in their ability to multiplex and quantify multiple peptides simultaneously, allowing for more robust, composite marker panels that capture different aspects of biology or disease processes. We have thus expanded the discussion accordingly. Here, a big factor is the isoform diversity in the human plasma proteome, which we now highlight much more extensively in the manuscript (see also our reply to reviewer #1 comment 7 on page 7)

In respect of multicenter studies, there is indeed one currently ongoing, using the OSPP internal standard (see comment 4 to Reviewer 2 on page 23); we also mention it in the discussion.

Minor but important.

1. The paper should not include cost-effectiveness as there is no cost associated with the kit yet, and this may not be the case. Please remove this from the manuscript.

We appreciate the reviewer's comment and agree that such claims should be made cautiously in the absence of an established commercial price for the OSPP kit. However, we believe it is important to highlight the design considerations that support potential cost-efficiency. Historically, the perceived high cost of using stable isotope-labeled (SIL) standards has been a barrier to their widespread adoption. To address this, OSPP was specifically designed with concentration-matched peptides and avoids including peptides that are not consistently detected for technical reasons (these add to the costs but usually are then discarded in the data analysis). Finally, the open nature of our standard allows everyone to order all or some of the peptides from any peptide biosynthesis company and thus create highly cost-effective standard panels in-house.

"The open design of our standard enables researchers to order part or all peptides from any peptide biosynthesis providers, allowing the creation of highly cost-effective internal standards in-house."

2. The protein full name is cardiac troponin T (not just troponin T, as there are several isoforms) and should also include cardiac troponin I, which is equally used clinically.

We thank the reviewer for this suggestion and have updated the respective name in the introduction.

3. Figure 5C is hard to read and should go in the online supplement, where it can be enlarged.

We thank the reviewer for the comment. We agree that the current version for reviewers is not easy to enlarge. We have provided a separate PDF of all figures.

4. There are a few typos that need to be fixed.

We thank the reviewer for the comments, we have proofread the version carefully, and we hope we spotted all the typos that were left.

References used in point by point reply

- Bernhardt, O.M. et al., 2012. Spectronaut A fast and efficient algorithm for MRM-like processing of data independent acquisition (SWATH-MS) data. *F1000Research*, 5. Available at: <https://doi.org/10.7490/F1000RESEARCH.1096450.1>.
- Bruheim, P., Kvitvang, H.F.N. & Villas-Boas, S.G., 2013. Stable isotope coded derivatizing reagents as internal standards in metabolite profiling. *Journal of chromatography A*, 1296, pp.196–203.
- Cox, J. et al., 2014. Accurate Proteome-wide Label-free Quantification by Delayed Normalization and Maximal Peptide Ratio Extraction, Termed MaxLFQ*. *Molecular & cellular proteomics: MCP*, 13(9), pp.2513–2526.
- Delongui, F. et al., 2013. Serum levels of high sensitive C reactive protein in healthy adults from southern Brazil: High sensitivity C reactive protein in adults. *Journal of clinical laboratory analysis*, 27(3), pp.207–210.
- Demichev, V. et al., 2021. A time-resolved proteomic and prognostic map of COVID-19. *Cell systems*, 12(8), pp.780–794.e7.
- Demichev, V. et al., 2019. DIA-NN: neural networks and interference correction enable deep proteome coverage in high throughput. *Nature methods*, 17(1), pp.41–44.
- Dierks, C. et al., 2025. Plasma proteomics reveals distinct signatures in occult and microfilaremic Loa loa infections. *The journal of infectious diseases*, p.jiaf344.
- Erlandsen, E.J. & Randers, E., 2000. Reference interval for serum C-reactive protein in healthy blood donors using the Dade Behring N Latex CRP mono assay. *Scandinavian journal of clinical and laboratory investigation*, 60(1), pp.37–43.
- Kurth, F. et al., 2020. Studying the pathophysiology of coronavirus disease 2019: a protocol for the Berlin prospective COVID-19 patient cohort (Pa-COVID-19). *Infection*, 48(4), pp.619–626.
- Lesur, A. et al., 2023. Quantification of 782 Plasma Peptides by Multiplexed Targeted Proteomics. *Journal of proteome research*, 22(6), pp.1630–1638.
- Messner, C.B. et al., 2021. Ultra-fast proteomics with Scanning SWATH. *Nature biotechnology*, 39(7), pp.846–854.
- Messner, C.B. et al., 2020. Ultra-High-Throughput Clinical Proteomics Reveals Classifiers of COVID-19 Infection. *Cell systems*, 11(1), pp.11–24.e4.
- Picotti, P. et al., 2009. Full dynamic range proteome analysis of *S. cerevisiae* by targeted proteomics. *Cell*, 138(4), pp.795–806.
- Thibeault, C. et al., 2021. Clinical and virological characteristics of hospitalised COVID-19 patients in a German tertiary care centre during the first wave of the SARS-CoV-2

pandemic: a prospective observational study. *Infection*, 49(4), pp.703–714.

Ulvik, A. et al., 2021. Quantifying precision loss in targeted metabolomics based on mass spectrometry and nonmatching internal standards. *Analytical chemistry*, 93(21), pp.7616–7624.

Wang, Z. et al., 2022. A multiplex protein panel assay for severity prediction and outcome prognosis in patients with COVID-19: An observational multi-cohort study. *eClinicalMedicine*, 49, p.101495.

World Health Organization, 2020. *WHO R&D Blueprint novel Coronavirus COVID-19 Therapeutic Trial Synopsis*, Available at: https://cdn.who.int/media/docs/default-source/blue-print/covid-19-therapeutic-trial-synopsis.pdf?sfvrsn=44b83344_1&download=true.

Point_by_point_reply_rev2

Reviewer #1 (Remarks to the Author):

Summary:

The authors have addressed most of the raised concerns and improved the text. I only have two important concerns remaining: a) the choice of some exemplary proteins to demonstrate the utility of the panels, particularly A2M; and b) the choice and design of statistical models. I believe the authors are well equipped to address these. Another important suggestion, though not one that should determine acceptance, concerns the supplementary tables – better descriptions and/or text references would benefit readers.

Specific comments :

1. Figure 3b: Please note that PCA is not a method for clustering, but a method for dimensionality reduction. It should not be referred to as unsupervised clustering, as done so in figure legends.

Ellipses are not confusing. Using ellipses with confidence intervals (CI) to show the spread and covariance structure of each group in a geometric space is excellent (95% CI preferred than 80%), but it's important to explain which method was used to calculate those areas. This information should be specified in the methods section.

We thank the reviewer for this helpful input.

i) We apologize for the confusion about the description of the PCAs, and have revised the figure 3b legend accordingly, it now reads

“Principal component analysis (PCA) of OSPP-normalized precursor quantities across samples. Each point represents a sample, colored by disease severity (as indicated in the legend), plotted along the first two principal components (PC1 and PC2), which explain 31.2% and 13.5% of the total variance, respectively. The overlaid arrows represent loadings of selected peptides/proteins, indicating their contribution to the principal components.”

ii) For Figure 3b, we had removed the 80% ellipses that were present in our original figure as part of the revision, based on the then-received reviewer feedback. We agree they might not be confusing to all readers, but they are no longer necessary, as the separation between the groups is now visualized with the loading plot.

iii) For Figure 5c, we agree with the reviewer that using a 95% confidence interval can provide a clearer representation of the subtle differences between platforms. Accordingly, we have updated the plot to display 95% CIs. We have also specified in the Methods section which package was used and how the ellipses were generated, as requested.

2. Figures 3c and 3d: The issue with C3 and CST3 is not because they are understudied. On the contrary, if the reviewers screen the literature, they will find many studies that show inconsistent association with COVID-19 status. Shouldn't in the context of this study, the

utility of the panel be demonstrated with regards to well-known changes in plasma protein levels in COVID-19 patients.

We thank the reviewer for this comment. We agree that we can illustrate the usability of the OSPP with other proteins than C3 and CST3, as the reviewer correctly states, that these two proteins are not COVID19 severity responsive across all studies. In order to address this comment, we now added additional examples of commonly reported COVID-19 response proteins which are covered by our standard panel, namely APOA2, ORM1 and SERPINA3 (Supplementary Figure 4, Ref 41,48-52). We also highlight now specifically the situation of cohort-dependency of the COVID-19 response of C3 and CST3 (Ref 61,62). Now the revised manuscript reads:

“Consistent with previous observations, most of the apolipoproteins classically associated with HDLs were less abundant in COVID-19 HDL particles, including APOA-II⁴⁸. In contrast, well-known key acute-phase plasma proteins such as Orosomucoid 1 protein (ORM1)⁴⁹ and Serpin A3 Protein (SERPINA3)⁵⁰ were increased depend on the severity of COVID-19, in agreement with our previous studies and other reports^{41,51,52} (Supplementary Figure 4).”

“Notably, some proteins such as C3 or CST3 are increasing in the presented cohort but not across all COVID-19 studies^{61,62}. These differences likely reflect cohort-specific effects.”

3. Figure 3 and text. On several occasions the authors write that they have performed ordinal linear regression, whereas under methods they describe ordinal logistic regression. These two methods are different and imply different assumptions. Please recheck your code to ensure that you have used ordinal logistic regression and correct the text. Related to this – does ordinal logistic regression require proportionality in the plasma protein change linked to COVID-19 severity?

We thank the reviewer for the helpful comment. We had used ordinal logistic regression. It is correct that the method assumes proportionality (across severity thresholds). On reflection, one can't be sure this assumption is always met. To avoid this assumption, we thus now use the Kendall's Tau trend test for both WHO grade and severity-based analyses. The corresponding P values in boxplots have been updated accordingly, and the updated results and details are provided in the revised Methods, Supplementary Tables, and Figures. Conclusions remain unchanged, but in this way we improve consistency in the use of statistics, and avoid potential pitfalls of ordinal logistic regression.

4. To my previous question about how consistent the association between A2M and COVID-19 is based on published evidence, the authors respond that there was a decline in severe COVID-19 in previous works and refer to references 49 and 50. Based on the text under results, I assume that the authors refer to references 48 and 49, since reference 50 is not related to the question. However, reference 48 is an old review from 2021 that concludes that “analysis of A2M protein levels in plasma samples from patients with COVID-19 revealed no significant differences or correlations with other disease parameters”. Reference 49 is a transcriptomics study that finds decrease in A2M transcript levels in peripheral blood mononuclear cells. Perhaps the question was not specific enough to imply that the evidence

in question is related to plasma proteomics data. The authors should still explain why A2M is highlighted as an exemplary protein to demonstrate the panel's utility for COVID-19 severity. I would further recommend evaluating the published literature on the relevance of this protein and reconsider highlighting more consistent proteins. In addition to the inconsistent published evidence, the authors highlight that A2M, an (moderate) acute phase protein decreases in severe COVID-19 patients. It is counterintuitive in a disease that has high systemic inflammation.

We agree that we can use other proteins than A2M to illustrate cross-platform normalization using the OSPP, and apologize if the references we have cited for this protein created confusion. To address the reviewer's comment, we have now removed the A2M peptide data and now demonstrate cross-platform normalization using another peptide (SPELQAEAK from APOA2), which is a well-known dysregulated protein in COVID-19 (Ref 53-55). We have also updated the references (previous Ref 48-50) to include recent studies reporting altered APOA2 levels in COVID-19 patients. Conclusions remain unchanged.

5. Adding a variance partitioning analysis is very useful. However, the supplementary table was poorly described. The authors need to clarify why the variance partitioning was performed per peptide and not on the entire dataset if they wanted to summarise the effect of biological signal vs. technical bias. Furthermore, the model is too simplistic to include only severity and MS as variables. The authors should consider including age, sex, site, sample type, and other potential confounders, and/or explain why other factors have not been included in the model. The authors then write under methods that a linear mixed-effects model was applied, but this makes the results unreliable since severity is on an ordinal scale, and not a proportional one (see question above). The authors should reconsider their model choice and use a model that is fitting for the used variables or explain how this model is applicable to the dataset. Last, the authors write that "missing values are imputed with zero" – please specify whether this refers to proteomics data or clinical data. In this setting the values should not be imputed for clinical data. For proteomic data, imputing with zero will severely reduce the variance in the data and a multiple imputation model should be considered.

We thank the reviewer for the insightful comment. The variance partition analysis was conducted at the peptide level, and we highlight this now better in the text.

We agree that a variance partition analysis could be used to discriminate against a number of biological confounders, including age, sex, site, sample type in study cohorts; however our exemplary COVID-19 patient cohort is not large enough to do this in any biological meaningful manner. Therefore, we revised the section, to highlight now better that our analysis focuses on reducing cross-platform variation for LC-MS proteomics using the OSPP as an internal standard. In order to address this comment of the reviewer, we have clarified this better in our text

"In a principal components analysis (PCA) of OSPP-normalized ratios for 187 peptides quantified across all platforms, the data showed strong cross-platform consistency. The first principal component (PC1, 22.7%) corresponded to the major variation associated with disease severities, while PC2 (10.5%) was mostly driven by technical variance between platforms (Figure 5c, left panel).

Visualization using 95% confidence ellipses highlighted the substantial overlap of data points across platforms, indicating minimal technical bias and consistent peptide quantification between different LC-MS platforms after OSPP normalization (Figure 5c, right panel). Variance partitioning analysis across peptides further confirmed that OSPP-based normalization substantially reduced technical variance attributed to cross-platform differences, from a median of 39.6% among all peptide quantities using median normalization down to 4.8% after OSPP normalization. (Supplementary Table 7 - variance partitioning across peptides)."

For imputation, in accordance with the reviewer's comment, we have replaced all zero imputation with k-nearest neighbor (KNN) imputation per peptide using the impute R package (v1.80.0). Details have been added to the Methods section accordingly and the modified result was updated in the respective supplementary table. Conclusions remain unchanged.

6. This also leads to me an additional point – the supplementary tables are not well organised and require improvement in structure. There are many sheets under a single Supplementary Table, and these should be referenced separately. E.g. the variance partitioning table was referenced as Supplementary Table 7, but it was on sheet 15 and the structure of the spreadsheets was not helpful in finding the right information.

We apologize if the numbering of our Supplementary material was confusing. We have carefully reviewed our table References, and now include detailed Table References to the main manuscript.

Reviewer #2 (Remarks to the Author):

The authors have satisfactorily addressed by comments and concerns in the revised version of their manuscript. I have no more comments.

Reviewer #3 (Remarks to the Author):

The authors have done a great job in addressing my concern. I particularly like the inclusion of the dynamic range figure. My only extremely minor but extremely important point is that the authors need to change gender to sex. Sex is the correct term as gender includes other factors than biological sex, such as social status, which is not what is captured in this data. Please make this change in Supplemental Figure 2 and throughout the manuscript. I missed this in my first review.

We thank the reviewer for the helpful suggestion. We have replaced all "gender" into "sex" in the revised manuscript and respective figures and supplementary figures.